

# On the calculation of single-scattering properties of frozen droplets and frozen droplet aggregates observed in deep convective clouds

Jeonggyu Kim[1], Sungmin Park[1], Greg M. McFarquhar[2,3], Anthony J. Baran[4,5], Joo Wan Cha[6], Kyoungmi Lee[6], Seoung Soo Lee[7], Chang Hoon Jung[8], Kyo-Sun Sunny Lim[9], Junshik Um[1,10,11]

[1]BK21 School of Earth and Environmental Systems, Pusan National University, Busan, Republic of Korea

[2]Cooperative Institute for Severe and High Impact Weather Research and Operations, University of Oklahoma, Norman, Oklahoma, USA

[3]School of Meteorology, University of Oklahoma, Norman, Oklahoma, USA

[4]School of Physics, Astronomy, Mathematics, University of Hertfordshire, Hatfield, UK

[5]Department of Observation-Based Research, Met Office, Devon, Exeter, UK

[6]Meteorological Applied Research Department, National Institute of Meteorological Sciences, Seogwipo, Republic of Korea

[7]Earth System Science Interdisciplinary Center, University of Maryland, Maryland, USA

[8]Department of Health Management, Kyungin Women's University, Incheon, Republic of Korea

[9]Department of Atmospheric Sciences, Center for Atmospheric Remote Sensing, Kyungpook National University, Daegu,

Republic of Korea.

[10]Department of Atmospheric Sciences, Pusan National University, Busan, Republic of Korea

[11]Institute of Environmental Studies, Pusan National University, Busan, Republic of Korea

*Correspondence to*: Junshik Um (jjunum@pusan.ac.kr)

**Abstract.** During multiple field campaigns, small quasi-spherical ice crystals, commonly referred to as frozen droplets (FDs), and their aggregates (frozen droplet aggregates (FDAs)), have been identified as the predominant habits in the upper regions of deep convective clouds (DCCs) and their associated anvils. These findings highlight the significance of FDs and FDAs for understanding the microphysics and radiative properties of DCCs. Despite the prevalence of FDs and FDAs at the tops of DCCs where they directly contribute to cloud radiative forcing, the detailed single-scattering properties (e.g., scattering-phase

function $P_{11}$ and asymmetry parameter $g$) of FDs and FDAs remain highly uncertain. This uncertainty is mainly due to insufficient in situ measurements and the resolution of cloud probes, which hinder the development of idealized shape models for FDs and FDAs. In this study, two shape models, the Gaussian random sphere (GS) and droxtal (DX), are proposed as possible representations for the shapes of in-situ measured FDs and FDAs. A total of 120 individual models of GSs and 129 models of DXs were generated by varying their shapes. Furthermore, by attaching these individual models in both

homogeneous or heterogeneous manners, three different types and a total of 315 models of FDAs were created: (1) aggregates of GSs; (2) aggregates of DXs; and (3) combinations of GSs and DXs which are called habit mixtures (HMs). The $P_{11}$ and $g$ of the developed models were calculated using a geometric optics method at a wavelength of 0.80 μm and then compared with those obtained using a Polar Nephelometer (PN) during the CIRCLE−2 field campaign to assess the models. Both individual component ice crystals (i.e., either GS or DX) and homogeneous component aggregates (i.e., either aggregates of GSs or

aggregates of DXs) showed substantial differences compared with the PN measurements, whereas the $P_{11}$ of the HMs was found to match most accurately the in situ measured $P_{11}$, reducing the differences to 0.87%, 0.88%, and 5.37% in the forward, lateral, and backward scattering regions, respectively. The $g$ of the HMs was found to be 0.80 which falls within the range of the PN measurement (0.78 ± 0.04). The root mean square error for the HM was minimized to a value of 0.0427. It was shown that the novel HMs developed in this study demonstrated better performance than in previous research where HMs were




developed indirectly by weighting the calculated $P_{11}$ of shape models to interpret in situ measurement. The result of this study
carries important implications for enhancing the calculation of single-scattering properties of DCCs.

**1 Introduction**

Deep convective clouds significantly influence the Earth's energy budget, hydrological cycle, and climate system (Jensen et
al., 1996; de Reus et al., 2009; Frey et al., 2011; Gayet et al., 2012; Raymond and Blyth, 2016). Characterized by intense

updrafts and towering vertical structures, these clouds play a crucial role in transporting heat, moisture, and momentum
throughout the troposphere (Houze, 2014; Lee et al., 2019), as well as injecting water vapor into the stratosphere (Dauhut and
Hohenegger, 2022). Vertical motions lift moist air, resulting in the formation of ice crystals and subsequent precipitation
(Andreae et al., 2004). Their radiative properties are crucial, affecting the Earth's radiative balance, through solar and terrestrial
radiation interactions. Understanding these properties is vital for climate models and feedback mechanisms. Furthermore, deep

convective clouds contribute to severe weather events, like thunderstorms, heavy rainfall, and lightning, impacting weather
prediction and societal concerns (Williams, 2018). Therefore, investigating their radiative properties advances our
comprehension of their atmospheric and climatic role, enhancing climate projections and weather forecasting.

Limited in situ observations have been made to characterize the microphysical and radiative properties of deep convective
clouds, especially in convective cores, due to safety concerns related to the vigorous updrafts (> 15 m/s), which prevent

penetrations of research aircraft into the cores. During the Cirrus Cloud Experiment (CIRCLE−2, Gayet et al., 2012), the Deep
Convective Clouds and Chemistry (DC3, Barth et al., 2015), and the CapeEx19 (Nairy, 2022) field campaigns, the
microphysical properties (e.g., size and habit distributions) of ice crystals at the upper levels of deep convective clouds were
measured. These upper regions of deep convective clouds were characterized by abundant quasi-spherical ice crystals, with
maximum dimensions ($D_{max}$) smaller than ~50 μm (Heymsfield and Sabin, 1989; Phillips et al., 2007, Lawson et al., 2010;

Järvinen et al., 2016), and their aggregates (Gayet et al., 2012; Baran et al., 2012; Stith et al., 2014; Järvinen et al., 2016; Um
et al., 2018). The presence of small quasi-spherical frozen droplets (FDs) has been attributed to homogeneous freezing of
supercooled droplets rapidly ascending in the updraft (Rosenfeld and Woodley, 2000; Gayet et al., 2012), whereas linear
"chain-like" shaped frozen droplet aggregates (FDAs) have been observed in environments with enhanced electric field (e.g.,
electrified thunderstorms) (Saunders and Wahab, 1975; Stith et al., 2002, 2004; Lawson et al., 2003; Connolly et al., 2005;

Um and McFarquhar, 2009; Pedernera and Ávila, 2018; Um et al., 2018).

Gayet et al. (2012) observed FDs and FDAs in the overshooting tops of a continental deep convective cloud at a temperature
of approximately −58 °C during CIRCLE−2 with unusually high concentrations of ice crystals up to 70 cm⁻³. A dense cloud
top exhibited a mean effective diameter of ~43 μm and a maximum particle size of approximately 300 μm, whereas the average
asymmetry parameter ($g$) was determined to be approximately 0.776 using a Polar Nephelometer (PN) (Crépel et al., 1997;

Gayet et al., 1997). Um et al. (2018) further investigated the morphological characteristics of FDs and FDAs using cloud
particle imager (CPI) data obtained during the DC3 field campaign. The CPI data were collected from the upper anvils of two
storms at altitudes between 12.0 and 12.4 km, at temperatures ($T$) ranging from −61 to −55 °C. It was revealed that FDs and
FDAs were the predominant habits, comprising 73.0 % (by number) and 46.3 % (by projected area) of the observed particles,
respectively. The average number (4.7 ± 5.0), size (31.79 ± 7.12 μm), and relative position of element FDs comprising the

FDAs were also determined (Um et al., 2018).

To quantify the radiative impacts of deep convective clouds, calculations of single-scattering properties of constituent ice
crystals are required, and idealized models representing realistic shapes of these constituent ice crystals should be developed.
The assumption of spherical shapes for the element FDs within FDAs was made to quantify the morphological characteristics
of FDs and FDAs by Um et al. (2018). This assumption is valid for determining the size (i.e., $D_{max}$) and the relative position



between FDs within FDAs, but it is not suitable for calculating the single-scattering properties (e.g., scattering-phase function $P_{11}$ and $g$) of non-spherically shaped FDs (Um and McFarquhar, 2011).

Even though a high resolution (i.e., 2.3 μm) CPI was used during the CIRCLE−2 (Gayet et al., 2012) and DC3 (Stith et al., 2014) campaigns to image FDs its resolution was not sufficiently high to fully resolve the three-dimensional morphological features (e.g., non-sphericity and surface roughness) of FDs. This limitation introduces uncertainties in the calculations of

single-scattering properties (Um and McFarquhar, 2011). In contrast, the FDs and FDAs imaged by the particle habit imaging and polar scattering (PHIPS, Abdelmonem et al., 2016) probe during the CapeEx19 campaign showed distinct non-spherical, plate-like shapes (Nairy, 2022). On the other hand, FDs captured in the vicinity of the convective core using the Ice Cryo−Encapsulation by Balloon (ICE−Ball) system showed quasi-spherical shapes with pronounced surface roughness (Magee et al., 2021). Laboratory−grown FDs and FDAs have also shown both quasi-spherical and non-spherical shapes of

FDs (Pedernera and Ávila, 2018).

Given the uncertainties surrounding the shapes of observed FDs and small ice crystals, several idealized shape models have been proposed to better represent the quasi-spherical nature of FDs and small ice crystals. These models include the Gaussian random sphere (Muinonen et al., 1996), droxtal (Yang et al., 2003), budding Bucky ball (Um et al., 2011), and Chebyshev particle (Mugnai and Wiscombe, 1980; McFarquhar et al., 2002; Um and McFarquhar, 2011; Baran et al., 2012). In particular,

two methods, the Gaussian random sphere and droxtal, produce shapes that closely resemble those observed in field campaigns and experiments (Thuman and Robinson, 1954; Othake, 1970; Yamazaki and Gonda, 1984; Pedernera and Ávila, 2018; Magee et al., 2021). The Gaussian random sphere is well-suited to describe FDs composed of roughened facets, while experiments conducted in a cold cloud chamber at temperatures below −40 ℃ suggest that the shapes of FDs more closely resemble those of droxtals (see Figs. 2−3 of Pedernera and Ávila, 2018). Although Baran et al. (2012) applied a weighted habit mixture model

of Chebyshev particles and hexagonal ice aggregates to calculate the single-scattering properties of FDAs, the development of idealized models specifically for FDAs and direct calculations of their single-scattering properties has yet to be thoroughly explored.

In this study, idealized shape models were developed using Gaussian random spheres and droxtals to represent the shapes of FDs and FDAs and the corresponding single-scattering properties were computed using a geometric optics method. The

results of these theoretical calculations are then compared with in situ measurements obtained during CIRCLE−2 to evaluate the developed models. The remainder of this paper is organized into the following sections: Section 2 outlines the development of shape models for FDs and FDAs based on in situ measurements. Section 3 details the theoretical methodology used to calculate the single-scattering properties of FDs and FDAs. Section 4 discusses the results, and Section 5 provides a summary and conclusion of this study.

## 110  2 Idealized models representing the shapes of frozen droplets and frozen droplet aggregates

To compute the single-scattering properties of FDs and FDAs, it is essential to have idealized shape models that closely replicate their natural form. In this study, shape models representing quasi-spherical FDs and FDAs were developed using Gaussian random spheres and droxtals based on the shapes of these particles observed during field campaigns and laboratory experiments. This section presents the geometrical description of the Gaussian random sphere and droxtal. Additionally, it

outlines the procedure used to construct shape models of FDAs.

### 2.1 Gaussian random sphere

The Gaussian random sphere is widely used to depict natural particles characterized by uneven surfaces, such as asteroids (Muinonen and Lagerros, 1998), desert dust particles (Nousiainen et al., 2003), and small ice crystals (Nousiainen and McFarquhar, 2004; Nousiainen et al., 2011). Magee et al. (2021) recently revealed previously undiscovered uneven and rough



surfaces due to the limited resolution of imaging probes. Therefore, the Gaussian random sphere is well-suitable for generating models to represent FDs because it is designed to represent uneven and rough surfaces. Here, idealized models of FDs, represented by Gaussian random spheres, were developed using the SIRIS software (Muinonen et al., 1996; Nousiainen et al., 2003). This software enables the generation of quasi-spherical shapes with randomly deformed surfaces through the utilization of Gaussian random sphere geometry along with several statistical parameters.

Two statistical parameters, the relative standard deviation of radius (σ) and correlation angle (Γ), were varied to generate a total of 120 Gaussian random spheres. 20 different Γ (i.e., from 0.01 to 0.20) were used to develop each case of Gaussian random spheres. Adjusting σ affects the radius vector (Υ) of the Gaussian random sphere and the radius vector is defined in spherical coordinates as

$$\Upsilon(\vartheta, \varphi) = \frac{a}{\sqrt{1+\sigma^2}} \exp[s(\vartheta, \varphi)]\,\hat{e_r} \text{ and} \tag{1}$$

$$s(\vartheta, \varphi) = \sum_{l=0}^{\infty}\sum_{m=-l}^{l} s_{lm} Y_{lm}(\vartheta, \varphi), \tag{2}$$

where $a$ denotes the mean radius, $Y_{lm}$ is the orthonormal spherical harmonics, and $s_{lm}$ is a Gaussian random variable generated with zero means (Muinonen et al., 2007). The $s$ and $\hat{e_r}$ is logradius and a unit vector pointing outward in the direction $(\vartheta, \varphi)$ in spherical coordinates, respectively. The Γ is correlation angle which is defined by

$$\Gamma = 2\arcsin\left(\frac{1}{2}l\right) \text{ and} \tag{3}$$

$$C_s(r) = exp(-\frac{2}{l^2}\sin^2\frac{1}{2}r), \tag{4}$$

where $C_s$ is the correlation function and $l$ is the correlation length (Nousiainen, 2002). Table 1 lists the values of these statistical parameters used to generate the Gaussian random spheres. Figure 1 shows example realizations of the geometric shapes of Gaussian random spheres with $D_{max}$= 30 µm. As Γ decreases while σ remains constant (i.e., from Fig. 1f to 1a), the shapes of Gaussian random spheres progressively deviate from a spherical form, assuming an increasingly spiky and appearance not like those of observed ice crystals. More detailed information on the SIRIS software and Gaussian random sphere geometry can be

found in Muinonen et al. (1996) and Nousiainen et al. (2003).

**Table 1. Statistical parameters, Γ and σ, used to develop six cases of FDs represented by Gaussian random spheres are shown.**

| case | 1 | 2 | 3 | 4 | 5 | 6 |
|------|-----|-----|-----|-----|-----|------|
| Γ | 10° | 20° | 30° | 60° | 90° | 180° |
| σ | 0.01 ~ 0.20 | | | | | |


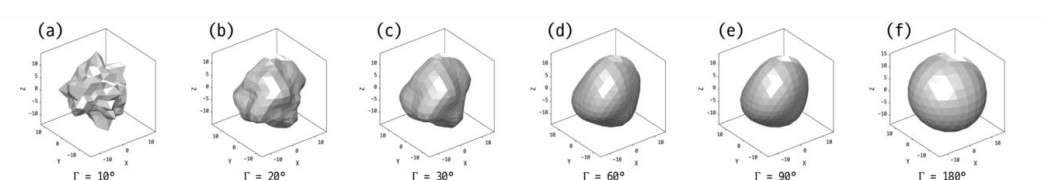

**Figure 1. Example realizations of the Gaussian random spheres based on six different Γ. The σ is 0.15 for all the cases presented**
**here and the values of Γ for each model are embedded at the bottom of each panel. The $D_{max}$ of Gaussian random spheres is identical (i.e., 30 µm) for all the cases.**



## 2.2 Droxtal

Thuman and Robinson (1954) researched Alaskan ice fog, collecting ice crystals using glass slides. They discovered unusual, small ice particles that exhibited characteristics of both droplets and crystals at temperatures less than –35 °C. These particles
were more prevalent than the well-formed hexagonal columns and plates typically expected. The researchers named these unique ice particles as droxtals. Later, Othake (1970) identified droxtals as 14 or 20−faced polyhedral crystals, noting that such particles are no longer considered unusual in ice fog conditions. Yamazaki and Gonda (1984) advanced this understanding by demonstrating the growth of a 20−faceted ice crystal from a frozen droplet through their experiments (see their Fig.1). The droxtals were used to calculate single-scattering properties of small quasi-ice crystals observed in cirrus clouds using the finite-
difference time domain method (Yang et al., 2003) and the improved geometric optics method (Zhang et al., 2004).

For $T < –40$ °C, cloud droplets freeze so rapidly that they do not have sufficient time to develop into typical shapes, such as hexagonal columns (Ohtake, 1970). Recent laboratory results reported that FDs observed at temperatures less than −40 °C bear a resemblance to the shape of droxtals (Pedernera and Ávila, 2018). Considering the aforementioned observations and experiments, it is plausible to assume the droxtal geometry as a possible candidate representing the shape of small and quasi-
spherical ice crystals which are observed in the upper anvil of DCCs.

Figure 2 shows the geometrical configuration of a droxtal. Two angular parameters $\theta_1$ and $\theta_2$, in conjunction with the radius ($R$) of a circumscribing sphere, determine the geometry of the droxtal. The relationships of these parameters are given by:

$$a_1 = R sin\theta_1 \quad \text{and} \quad a_2 = R sin\theta_2 \quad \text{and} \tag{5}$$

$$L_1 = R cos\theta_1 \quad \text{and} \quad L_2 = R cos\theta_2 \quad \text{and} \tag{6}$$

where $a_1$ and $a_2$ are connected with the area of hexagonal faces of the droxtal. In Eq. (6), $L_1$ and $L_2$ are related to the area of rectangular and trapezoidal faces, respectively. With a total of 20 faces, a single droxtal features 12 trapezoidal faces (e.g., EFF'E' in Fig. 2a), 6 rectangular faces (e.g., E'F'F''E'' in Fig. 2a), and 2 hexagonal faces (e.g., ABCDEF in Fig. 2a). The single droxtal, specified by $\theta_1 = 32.35°$ and $\theta_2 = 71.81°$, exhibits maximum sphericity. Based on this model representing maximum sphericity, modifications were made to generate the idealized droxtal models. By adjusting either $\theta_1$ or $\theta_2$ at one-
degree intervals, a total of 127 idealized droxtal models with $D_{max} = 30$ μm were developed. Four example realizations of the droxtal are illustrated in Fig. 3. Figure 3a and 3b correspond to the case where $\theta_1$ varies, while $\theta_2$ is fixed at 71.81°. Conversely, Figs. 3c and 3d show the cases where $\theta_2$ is varied, while $\theta_1$ is held constant at 32.35°. The corresponding values of $\theta_1$ and $\theta_2$ are shown at the bottom of each panel. Table 2 gives the values and ranges of the angular parameters for each droxtal case developed in this study.


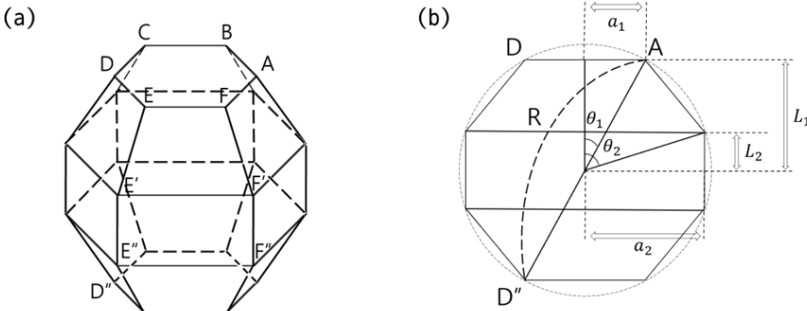

**Figure 2. Geometrical configuration of a droxtal. The $\overline{AD''}$ in (b) corresponds to the maximum dimension ($D_{max}$). Adjusting the two angles, $\theta_1$ and $\theta_2$, influences the area of hexagonal, rectangular, and trapezoidal faces of droxtal.**






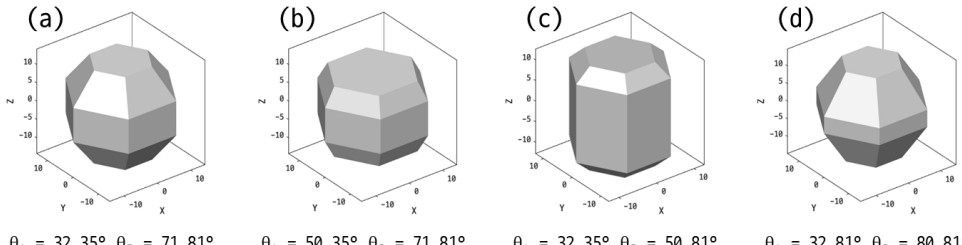

**Figure 3. Example realizations of droxtal models based on the four cases shape statistics. The $D_{max}$ for each droxtal model is 30 μm. Corresponding statistical parameters, $\theta_1$ and $\theta_2$, are shown at the bottom of each panel.**


**Table 2. Statistical parameters, $\theta_1$ and $\theta_2$, used to develop four cases of FDs represented by droxtals are shown.**

| case | 1 | 2 | 3 | 4 |
|------|---|---|---|---|
| $\theta_1$ | 1.35° ~ 32.35° | 33.35° ~ 71.35° | 32.35° | 32.35° |
| $\theta_2$ | 71.81° | 71.81° | 32.81° ~ 71.81° | 72.81° ~ 89.81° |


### 2.3 Frozen droplet aggregates

Linearly chained FDAs are one of the distinct characteristics of continental DCCs, which may be produced by high electric fields that exist within the clouds (Saunders and Wahab, 1975; Stith et al., 2002, 2004, 2014; Connolly et al., 2005; Gayet et al., 2012; Järvinen et al., 2016; Um et al., 2018). According to an analysis on the morphological properties of FDAs by Um et

al. (2018), on average, FDAs consisted of 4.7 ± 5.0 individual FDs, with approximately 90 % of the measured FDAs being composed of 2 to 10 individual FDs.

In this section, idealized models representing the linearly-chained and complex shapes of FDAs are developed. To this end, individual models of FDs (i.e., either Gaussian random spheres or droxtals) were distributed in three-dimensional space with random orientations, reflecting the natural tendency of ice crystals in the atmosphere to have no preferred orientations.

Additionally the following assumptions were made to develop the idealized models of FDAs.

-     The shape of all FDs composing a FDA model is identical.
-     No overlap exists between the constituents of FDA models.
-     The maximum number of contact points between the constituents of a FDA model is two.

The aggregation index (*AI*) is used to describe the shape of FDAs in three-dimensional space. The *AI* has been used for

analyzing the impacts of three-dimensional shapes of aggregates of ice crystals, such as bullet rosettes and plates, on their corresponding scattering properties (Um and McFarquhar, 2009; Um et al., 2018). The *AI* is defined as

$$AI = \frac{\Sigma_i^n \Sigma_j^n D_{ij}}{MAX(\Sigma_i^n \Sigma_j^n D_{ij})}, \tag{7}$$

where $D_{ij}$ is the distance between the center of the circumscribing circle of frozen droplet *i* and that of frozen droplet *j*. The *AI* is calculated only for the cases where $n \geq 3$ (Um and McFarquhar, 2009; Um et al., 2018). As the *AI* value increases, the shape



of FDAs becomes more similar to that of a linearly-chained structure. A total of 270 idealized FDAs models were generated,
varying the number of constituent FDs (i.e., *n* ranging 2 to 10). Figure 4 shows examples of aggregates of 6 Gaussian random
spheres and aggregates of 4 droxtals.

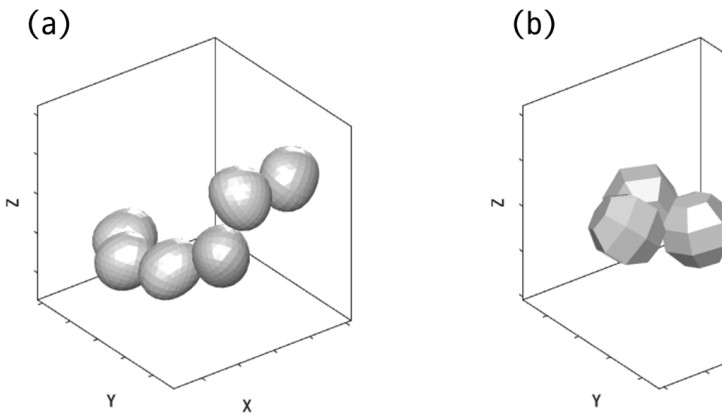

**Figure 4. Examples of newly developed models for FDAs. (a) FDAs represented by aggregates of six Gaussian random spheres and,**
**(b) FDAs represented by aggregates of four droxtals.**

**3 Geometric optic method for calculating single-scattering properties**

The $P_{11}$ and *g* are single-scattering properties of great interest in both remote sensing and climate models studies. The $P_{11}$ is
the first element of the phase matrix (P) that describes the scattering intensity of radiation when the incident light is unpolarized.
(Bohren and Huffman, 1983). The *g*, which is defined as the cosine-weighted normalized $P_{11}$, provides a measure for assessing
asymmetry in the forward scattering region of the phase function. The *g* takes on values between -1 and 1 depending on the
direction of scattered energy. The *g* equals 1 when all radiation is scattered into the forward hemisphere, and -1 when all
radiation is scattered in the backward hemisphere. When *g* equals 0, this indicates isotropic or hemispherically symmetric
scattering. The normalized $P_{11}$ and *g* are defined as

$$P_{11} = \frac{1}{4\pi} \int_{4\pi} P_{11} \, d\Omega, \tag{8}$$

$$g = \frac{1}{4\pi} \int_{4\pi} P_{11} \cos\theta d\Omega, \tag{9}$$

where $\Omega$ and $\theta$ is the solid angle and scattering angle, respectively (Bohren and Huffman, 1983).

The geometric optics method (GOM), also known as ray optics or the ray-tracing method, is a widely used approximation
technique that calculates single-scattering properties (e.g., $P_{11}$ and *g*) of atmospheric ice crystals (e.g., Yang and Liou, 1995;
Macke et al., 1996; Konoshonkin et al., 2015). Furthermore, various methods derived from the GOM, such as improved
geometric optics methods (IGOM, Yang and Liou, 1998; Havemann et al., 2002), ray tracing with diffraction on facets (RTDF,
Hesse et al., 2008, 2009), and geometrical-optics-integral-equation (GOIE, Ishimoto et al., 2012) have been developed and are
in use. The GOM is applicable when the size parameter ($\chi = \frac{2\pi r}{\lambda}$) of scatterers is significantly larger than the wavelength ($\lambda$)
of incident light (i.e., $\chi \gg 1$), but the lower size limit of the applicability of conventional GOM is not well-defined, and it
depends on the morphology of scatterer (Um and McFarquhar, 2015). For the calculation of single-scattering properties of





FDs and FDAs, the modified version of the geometric ray-tracing code (Macke et al., 1996), which includes parallelized computation, was used (Um and McFarquhar, 2009).

For the simulations presented here, 72,000 randomly chosen orientations and 1,000 incoming rays per orientation were used
to calculate single-scattering properties at $\lambda = 0.80$ μm. The refractive index of ice at this $\lambda$ is $1.3049 + i1.34 \times 10^{-7}$ (Warren and Brandt, 2008), where $i$ is the imaginary part, and a scattering angle resolution of 0.25° was employed. The results of simulations are compared with the single-scattering properties measured by a PN during CIRCLE−2 to determine the best-fit model that most accurately matches the observations. The best-fit model minimizing the root mean square error (RMSE), $X_{RMSE}$, was determined as best-fit model. $X_{RMSE}$ is defined as

$$X_{RMSE} = \sqrt{\frac{1}{N}\left\{\sum_{i=1}^{i=32} X_i^2\right\}}, \tag{10}$$

where $X_i$ represents the log difference between the measured average $P_{11}$ and computed $P_{11}$. The $i$ ($i = 1, 2, …, 32$) is the number of scattering angles measured by the PN instrument.

## 4 Calculation results and comparison with in situ measurements

To determine the ice crystal model that best matches the $P_{11}$ and $g$ of FDAs observed by the PN during CIRCLE−2, the single-scattering properties of individual crystals (i.e., either Gaussian random spheres or droxtals) with varying shapes and those of their aggregates (i.e., either aggregates of Gaussian random spheres or aggregates of droxtals) with varying 3D morphologies (e.g., $AI$) were calculated. Initially, the single-scattering properties of the individual component ice crystals were calculated and then compared with the PN measurements in Section 4.1. A total of 120 Gaussian random spheres (discussed in Sect. 4.1.1)
and 129 droxtals (discussed in Sect. 4.1.2) were used for this purpose. This step is crucial to verify whether a constituent crystal can represent the single-scattering properties of their aggregates, as previous studies (Um and McFarquhar, 2007; 2009) have demonstrated similarities between the single-scattering properties of aggregate crystals and their component crystals. Subsequently, the calculated single-scattering properties of homogeneous component aggregates (i.e., either aggregates of Gaussian random spheres (Sect. 4.2.1) or aggregates of droxtals (Sect. 4.2.2)) were compared with those observed by the PN
(Sect. 4.2). Finally, models for heterogeneous component aggregates, which are mixtures of Gaussian random spheres and droxtals, were developed in Section 4.3.

### 4.1 Single frozen droplets

#### 4.1.1 Gaussian random sphere

Figure 5a illustrates the comparison between the $P_{11}$ of 120 single FDs models, represented by Gaussian random spheres, and
data obtained during CIRCLE−2 (i.e., gray shading area). The $P_{11}$ was divided into three different scattering regions: forward scattering (0° to 60°), lateral scattering (60° to 120°), and backward scattering (120° to 180°) angles, to analyze the relative contribution to each region. Single models of Gaussian random spheres showed average differences of 10.63%, 47.28%, and 32.19% in the forward, lateral, and backward scattering regions, respectively compared with the PN measurements. In particular, the energy scattered into the lateral and backward directions exhibited notable differences compared to that scattered
into the forward region. This discrepancy is attributed to the typical characteristic of quasi-spherical ice crystals, low lateral-scattering, as identified by Mishchenko and Travis (1998). Additionally, unlike other non-spherical ice crystals (e.g., hexagonal columns and droxtals), the $P_{11}$ of the Gaussian random sphere did not show sharp peaks in the forward scattering region. The average $g$ was $0.83 \pm 0.05$ which falls outside the measurement range of the PN (i.e., $0.78 \pm 0.04$). However, it was shown that the Gaussian random spheres representing spiky forms, as illustrated in case (a), (b), and (c) of Fig. 1, have $g$ values with



uncertainty ranges of 0.77 ± 0.04, 0.80 ± 0.04, and 0.81 ± 0.04, respectively, which are close to those of the PN (indicated by
blue dashed line in Fig. 5b). These results, as clearly depicted in Fig. 5b, show that the spherical shape models scatter more
intensity into the forward scattering region, consequently leading to an increased $g$.

To address the discrepancy in the lateral scattering observed in spherical cases of Gaussian random spheres (i.e., (d), (e),
and (f) of Fig. 1), additional simulations using the distortion parameter ($t$) were conducted. Using $t$ is a statistical method
reflecting the influence of distorted faces, surface roughness, or inclusions of ice crystals on the single-scattering properties.
It involves tilting the path of the reflected and refracted ray randomly during the simulation, around its original direction. The
zenith and azimuth tilt angles are randomly selected with an equal distribution between 0 and $\theta_t^{max}$ ($0 \leq \theta_t^{max} \leq 2\pi$). The
degree of distortion is defined by the relation: $t = \theta_t^{max}/90°$ (Macke et al., 1996).

Figure 6 shows the $P_{11}$ of Gaussian random spheres when assuming varying $t$ values: 0.0 (indicating no distortion), 0.1, 0.2,
and 0.3. Each black-colored line corresponds to the $P_{11}$, with different values of $t$, exhibiting the smallest RMSE when
compared with the $P_{11}$ measured by the PN (indicated by filled red circles in Fig. 6). A single Gaussian random sphere with $t$
= 0.3 was the best-fit model minimizing the RMSE to a value of 0.1243. The differences in forward, lateral, and backward
scattering for the best-fit model were 6.89%, 39.32%, and 20.16%, respectively. The $g$ was calculated to be 0.76 which falls
within the measurement range of the PN.

The $t$ was applied for the purpose of reducing the difference in the lateral scattering region caused by general features of
quasi-spherical ice crystals (i.e., low lateral scattering), however, it was not sufficient to minimize the RMSE value. The
differences in the $P_{11}$ and $g$ between the Gaussian random sphere models and in situ measurements, across distortion parameter
values, are summarized in Tables 3 and 4.

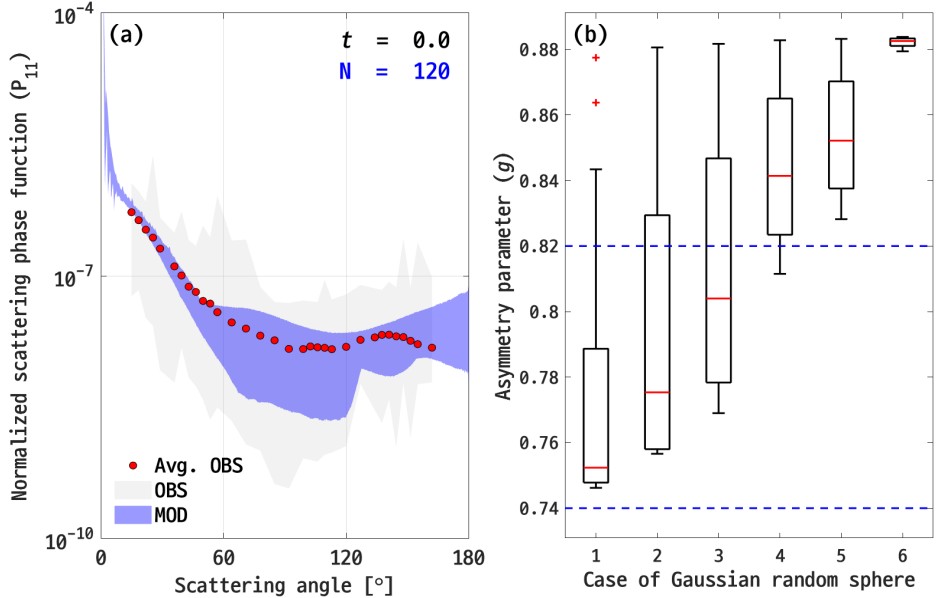

**Figure 5. (a) A comparison of the computed $P_{11}$ for single Gaussian random sphere models (blue shaded area) against $P_{11}$ obtained during CIRCLE−2 (red filled circles). The gray-shaded area represents the full range of $P_{11}$ measurements of ice crystals obtained during CIRCLE−2. (b) The computed $g$ for each case of individual Gaussian random spheres. The blue dashed lines indicate the uncertainty range for $g$ of PN measurement (i.e., 0.78 ± 0.04). The median values for each case of the single Gaussian random sphere model are plotted as red solid lines in the box plot.**



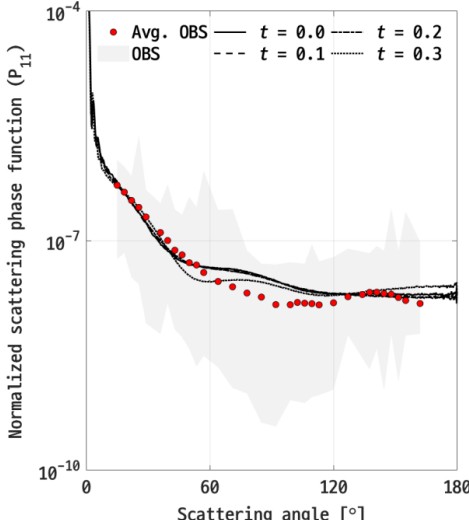

**Figure 6. A comparison of the average in situ measured $P_{11}$ (red filled circles) with the $P_{11}$ of a single Gaussian random sphere. Each black line corresponds to the best-fit model assuming $t$ values from 0.0 to 0.3. The gray-shaded area represents the full range of $P_{11}$ measurements obtained during CIRCLE−2.**

### 4.1.2 Droxtals

Figure 7a shows a comparison of the $P_{11}$ of droxtals with the in situ measured $P_{11}$. Due to their faceted structure, the droxtals have $P_{11}$ characterized by several sharp peaks in the forward scattering directions (Zhang et al., 2004; Um et al., 2011; Yang et al., 2013). The strong peaks are distinctly visible in the $P_{11}$ of droxtals in Fig. 7a and the most pronounced peak was found at 6.5°. These peaks indicate angles of minimal deviation, caused by the refraction of rays passing through two facets of a particle.

For the droxtals, the average differences between the droxtal models and in situ measurements in the forward, lateral, and backward scattering were 4.80%, 27.06%, and 15.66% respectively. The droxtals show a smaller difference in the lateral scattering direction compared to that of the Gaussian random sphere. The overall shape of the $P_{11}$ in the forward scattering region appears to deviate significantly from observations. However, these models still effectively simulate the total intensity scattered into the region, outperforming the Gaussian random spheres. Figure 7b illustrates the comparison of $g$ values of droxtals with the measurement range of the in-situ measurements (indicated by blue dashed line in Fig. 7b) using box plots. Droxtals representing the typical shapes (i.e., (a) in Fig. 3) and the shape most closely resembling columns (i.e., (c) in Fig. 3) have average $g$ of $0.81 \pm 0.01$ and $0.82 \pm 0.01$, respectively. These results fall within the upper measurement range of the PN measurements (i.e., $0.78 \pm 0.04$).

Figure 8 shows the $P_{11}$ of droxtals with the smallest RMSE for $t$ ranging from 0.0 to 0.3. Each $P_{11}$ of the droxtal is depicted using the black line, whereas the in situ measurements are represented by red filled circles. The $P_{11}$ of droxtals with higher $t$ exhibit smaller difference from the in situ measurements in the lateral and backward scattering region in contrast to the Gaussian random spheres. The disappearance of strong peaks in the forward scattering region, as the $t$ value increases from 0.0 to 0.3, leads to improved agreement in the forward scattering region. Similar to the case with a single Gaussian random sphere, a single droxtal with $t = 0.3$ minimized the RMSE to 0.0806. The calculated differences between the $P_{11}$ of the droxtal with $t = 0.3$ and the in situ measured $P_{11}$ were 2.49%, 3.48%, and 10.64% in the forward, lateral, and backward scattering regions, respectively. The $g$ was found to be 0.81 which is close to the upper boundary of the measurement range of PN. Tables




3 and 4 provide a summary of the average and standard deviation for the differences in the $P_{11}$ and $g$ between developed droxtal models and the in situ measurements at different $t$ values.


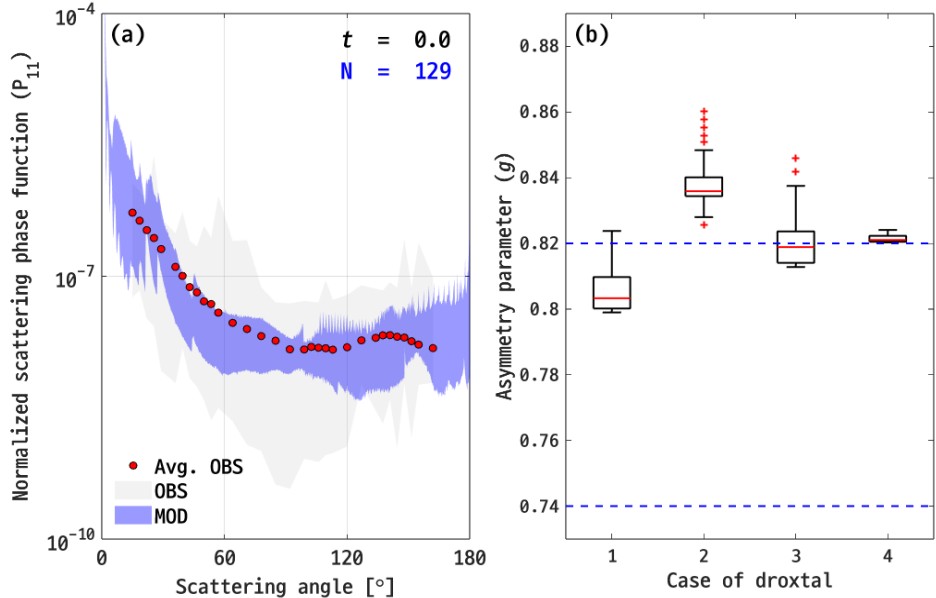

**Figure 7. Same as Fig. 5 but for the single droxtal models.**


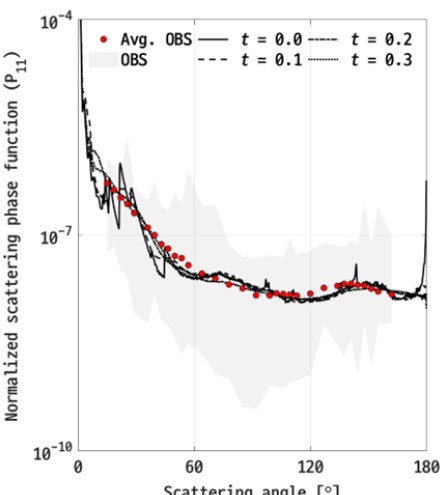

**Figure 8. Same as Fig.6 but for the single droxtal models.**






**Table 3.** Average differences and standard deviations in $P_{11}$ for shape models representing single FDs (either Gaussian random spheres (GS) or droxtals (DX)), across forward (FWD), lateral (LAT), and backward (BWD) scattering regions, with $t$ values ranging from 0.0 to 0.3.

| Difference in $P_{11}$ (%) | | $t = 0.0$ | $t = 0.1$ | $t = 0.2$ | $t = 0.3$ |
|---|---|---|---|---|---|
| | FWD | 10.63 ± 5.19 | 10.58 ± 5.23 | 10.44 ± 5.26 | 10.18 ± 5.24 |
| GS | LAT | 47.28 ± 27.67 | 46.91 ± 27.48 | 45.96 ± 26.73 | 44.47 ± 25.42 |
| | BWD | 32.19 ± 12.21 | 32.41 ± 12.16 | 32.78 ± 12.35 | 32.85 ± 13.36 |
| | FWD | 4.80 ± 1.90 | 4.85 ± 2.02 | 4.80 ± 2.10 | 5.63 ± 1.56 |
| DX | LAT | 27.06 ± 11.23 | 27.07 ± 11.53 | 26.74 ± 11.02 | 27.05 ± 11.29 |
| | BWD | 15.65 ± 11.09 | 15.38 ± 11.15 | 15.43 ± 11.72 | 14.76 ± 12.95 |


**Table 4.** Average differences and standard deviations in $g$ for shape models representing single FDs (either Gaussian random spheres (GS) or droxtals (DX)), across forward (FWD), lateral (LAT), and backward (BWD) scattering regions, with $t$ values ranging from 0.0 to 0.3.


| Difference in $g$ (%) | $t = 0.0$ | $t = 0.1$ | $t = 0.2$ | $t = 0.3$ |
|---|---|---|---|---|
| GS | 7.73 ± 4.71 | 7.70 ± 4.69 | 7.52 ± 4.57 | 7.18 ± 4.57 |
| DX | 6.01 ± 1.88 | 5.99 ± 1.86 | 5.81 ± 1.83 | 5.40 ± 1.78 |

### 4.2 Aggregates of single frozen droplets

Based on the results of the single-particle models discussed in Section 4.1, models of FDAs were constructed using the
Gaussian random sphere and droxtal, which provided the smallest RMSE in comparison to the in situ measurements. These
developed FDAs featured homogeneous components, with the component shapes of the FDAs varying and the number of
components in each model ranging from 2 to 10. For each particle count, 15 different FDAs models were generated (e.g.,
creating 15 models for 2-particle aggregates, 15 for 3-particle aggregates, etc.). This approach was applied to both the Gaussian
random spheres and droxtals, so that the total number of developed models amounts to 270.

As the observed FDAs have a myriad range of morphologies including "linearly-chained" or "compact" shapes (Gayet et
al., 2012; Stith et al., 2014; Um et al., 2018), diverse 3D morphological characteristics should be considered. The *AI* (discussed
in Sect. 2.3) was introduced for this purpose. Analysis on the *AI* of the constructed FDAs reveals that for models composed of
homogeneous Gaussian random spheres, the *AI* values range from a minimum of 0.5305 to a maximum of 0.9457, with a mean
value of 0.7567. For those composed of homogeneous droxtal components, the minimum, maximum, and mean *AI* values are
0.4224, 0.9738, and 0.7321, respectively.

### 4.2.1 Aggregates of Gaussian random spheres

Figure 9 shows the $P_{11}$ and $g$ at $\lambda = 0.80$ μm for aggregates of Gaussian random spheres. The aggregates of Gaussian random
spheres showed an average difference of 13.47%, 80.65%, and 11.06% from the in-situ measurements in the forward, lateral,
and backward scattering regions, respectively. Although these are minimized to 12.41%, 75.93%, and 6.83% with $t = 0.3$, the
discrepancies in the lateral scattering angles remain. One distinct feature is that the discrepancies in the lateral scattering




direction from in-situ measurements evident in the aggregate of Gaussian random sphere models were also observed for individual Gaussian random sphere models (see Fig. 5a and Fig. 6). These discrepancies remain or even become more pronounced due to the interactions among the components of FDAs. Figure 10 shows the variations of differences in the lateral scattering region of the $P_{11}$ from in-situ measurements with $t$ from 0.0 to 0.3. As shown in Fig. 10, as more components are
included in a single model of FDAs, the difference between the model and observations tends to increase.

Figure 9b shows how the calculated $g$ varies as a function of the number of attached Gaussian random spheres. An increase in the number of components of FDAs corresponds to a decrease in $g$. This is attributed to the fact that a higher number of chained FDs results in a longer path length for reflected and refracted rays, consequently leading to a decrease in scattered energy in the forward scattering region. Despite being close to the lower boundary of the measurement range of the PN, FDAs
composed of 2 or 3 Gaussian random spheres were found to simulate the $g$ values closer to those observed compared to those with more than 4 components.

Figure 11 shows the $P_{11}$ for aggregates of Gaussian random spheres that have the smallest RMSE with $t$ from 0.0 to 0.3 (i.e., black lines in Fig. 11). The best-fit model, found to be composed of four Gaussian random spheres of type 1 (i.e., (a) in Fig. 1), minimized the RMSE value to 0.1338 when applying $t = 0.3$. The differences between the $P_{11}$ of the best-fit model and the
in situ measured $P_{11}$ were 10.43%, 67.60%, and 0.27% in the forward, lateral, and backward scattering regions, respectively. The $g$ was 0.75 which is close to the lower boundary of the PN's uncertainty range. While the $P_{11}$ shows rather reasonable agreement with the in situ measurements in the forward scattering region, it does not accurately simulate the lateral and backward scattering regions of the measurements and notably, no broad peaks are evident at a scattering angle of approximately 140°.


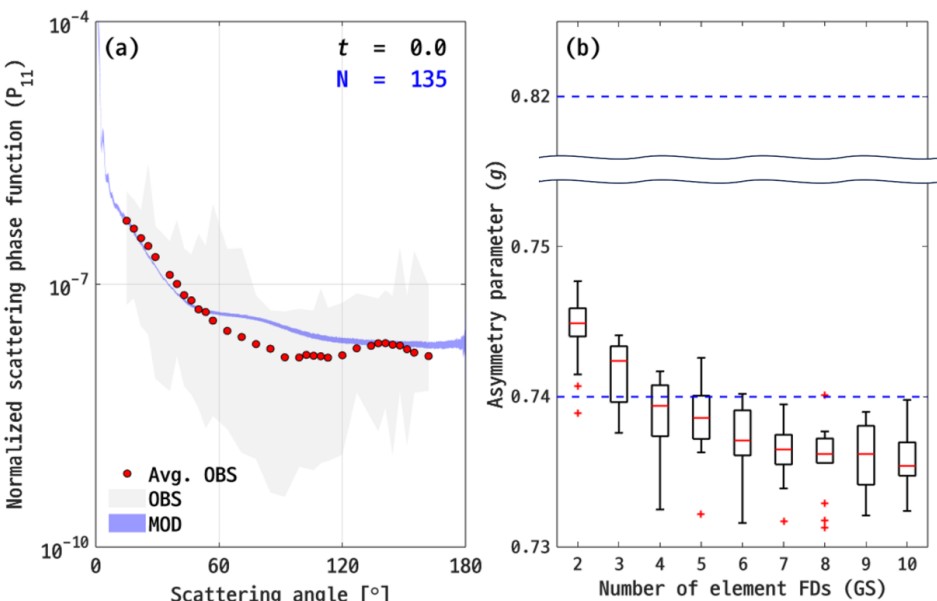

**Figure 9. (a) The calculated $P_{11}$ for FDAs represented by aggregates of Gaussian random spheres with distortion parameter $t = 0.0$ at $\lambda = 0.80$ μm. The gray-shaded area and red filled circles represent the full range of $P_{11}$ measurements obtained during CIRCLE−2,**
**respectively. The blue-shaded area indicates the computed $P_{11}$ for FDAs and N denotes the total number of aggregates of Gaussian random spheres developed in this study. (b) The variation in computed $g$ values as a function of the number of components. The red solid line and blue dashed line represent the median $g$ value for each component count and uncertainty range for the $g$ measured by PN, respectively.**






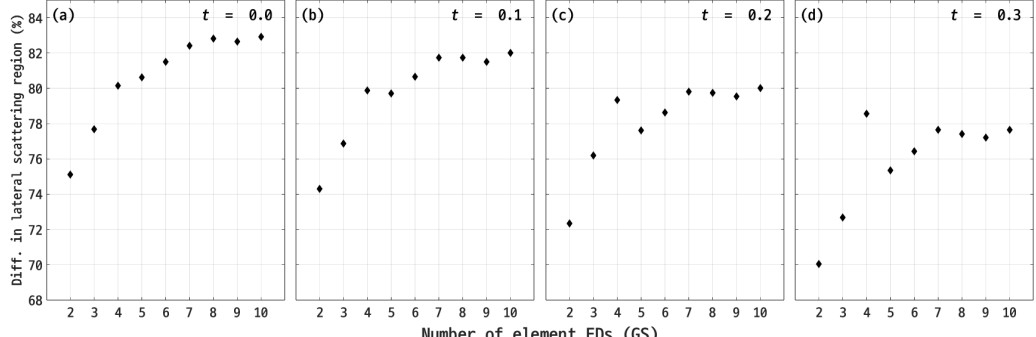

**Figure 10. The differences between the in situ measured $P_{11}$ and computed $P_{11}$ for the FDAs consisting of Gaussian random spheres. As the number of components increases, the differences between the model and observations increase.**


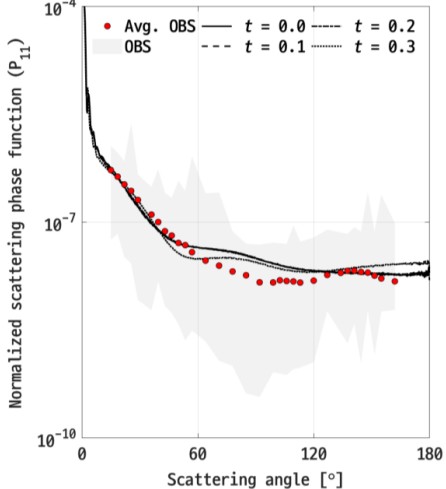

**Figure 11. A comparison of the $P_{11}$ computations for aggregates of Gaussian random spheres with the average $P_{11}$ obtained during CIRCLE−2. Each black line corresponds to the best-fit model assuming $t$ values from 0.0 to 0.3. The gray-shaded area represents the full range of $P_{11}$ measurements from CIRCLE−2.**


### 4.2.2 Aggregates of droxtals

Figure 12a illustrates a comparison of $P_{11}$ for aggregates of droxtals with the in situ measurements at $\lambda = 0.80$ μm. In the

forward scattering region, peaks are evident due to the faceted structure of droxtals as shown in Fig. 3. For droxtals with $t = 0.0$, the average differences in the forward, lateral, and backward scattering regions were 3.07%, 20.39%, and 11.27%, respectively. These differences are reduced to 1.02%, 11.84%, and 3.33% in the forward, lateral, and backward regions, respectively, when assuming $t = 0.3$.

Figure 12b presents the variation in the calculated $g$ as a function of the number of attached droxtals. The $g$ values of FDAs

represented by droxtals have a pattern similar to those of Gaussian random sphere aggregates, characterized by a decrease in





the $g$ values as the number of attached components increases (see Fig. 9b). The average differences from the in situ measurements and standard deviations for both $P_{11}$ and $g$ of droxtal aggregates, ranging from no distortion ($t = 0.0$) to $t = 0.3$, are summarized in Tables 5 and 6.

Figure 13 shows that the difference from in-situ measurements in lateral scattering for FDAs models represented by droxtals increases with the number of components. However, this difference slightly decreases as the $t$ increases from 0.0 to 0.3. In Figure 14, each black line corresponds to the calculated $P_{11}$ for aggregates of droxtals which provide best-fit to the measured $P_{11}$ with $t$ ranging from 0.0 to 0.3. The best-fit model consisted of two droxtals of type 3 (i.e., (c) of Fig. 3), which gave a minimum RMSE of 0.0550 with $t = 0.3$. The average differences of the best-fit model from in situ measurements were 1.19%, 12.43%, and 1.41% in the forward, lateral, and the backward scattering regions, respectively.

The best-fit model, consisting of aggregates of droxtals, significantly improves agreement with in situ measurements in the lateral scattering region, reducing the difference down to 12.43%. It closely simulates the measurements not only in the lateral scattering region but also in the backward regions, particularly capturing peaks around a 140° scattering angle. The computed $g$ for the best-fit model was 0.79 which falls within the measurement range of PN. Tables 5 and 6 summarize the average differences and standard deviations in $P_{11}$ and $g$ values between droxtal aggregates and in situ measurements, across $t$ values

ranging from 0.0 to 0.3.

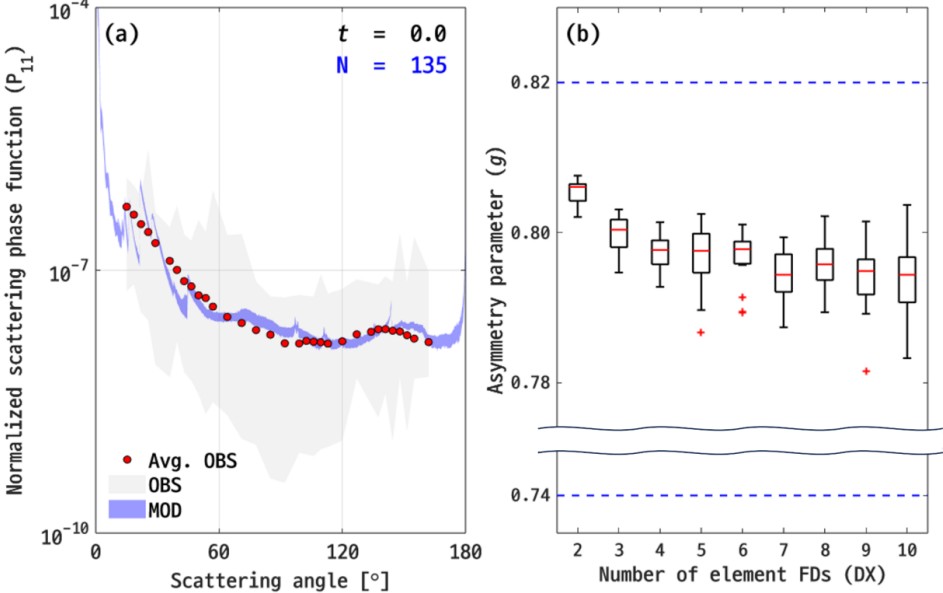

**Figure 12. Same as Fig. 9 but for the models of FDAs represented by aggregates of droxtals.**





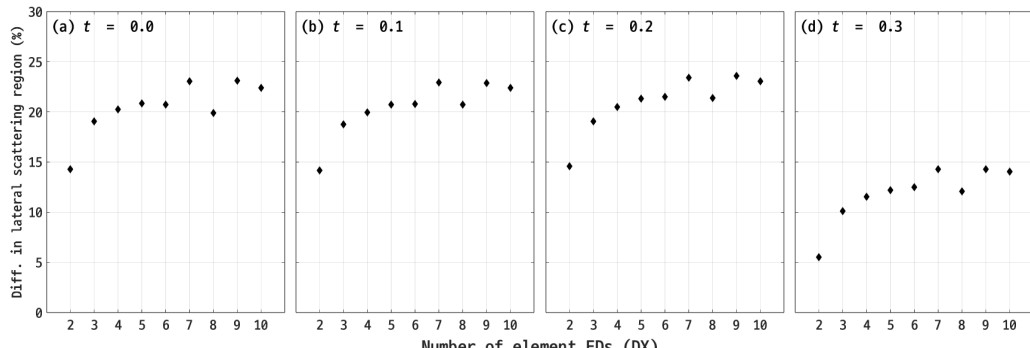

**Figure 13. Same as Fig. 10 but for the models of FDAs represented by aggregates of droxtals.**


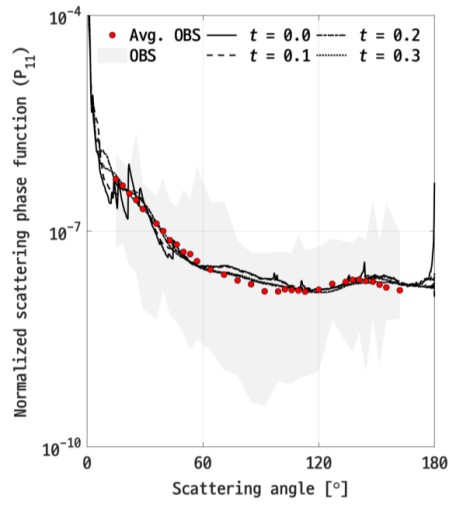

**Figure 14. Same as Fig.11 but for the FDAs represented by aggregates of droxtals.**


**Table 5. Same as Table 3 but for the FDAs models represented by aggregates of Gaussian random spheres (FDA_GS) or aggregates of droxtals (FDA_DX).**

| Difference in $P_{11}$ (%) | | $t = 0.0$ | $t = 0.1$ | $t = 0.2$ | $t = 0.3$ |
|---|---|---|---|---|---|
| | FWD | $13.47 \pm 0.74$ | $13.28 \pm 0.73$ | $12.90 \pm 0.72$ | $12.40 \pm 0.77$ |
| FDA_GS | LAT | $80.65 \pm 3.16$ | $79.83 \pm 3.13$ | $78.14 \pm 3.07$ | $75.89 \pm 3.31$ |
| | BWD | $11.06 \pm 2.65$ | $10.35 \pm 2.63$ | $8.91 \pm 2.57$ | $6.81 \pm 2.82$ |
| | FWD | $3.07 \pm 0.92$ | $3.10 \pm 0.91$ | $3.22 \pm 0.88$ | $1.02 \pm 0.71$ |
| FDA_DX | LAT | $20.39 \pm 4.20$ | $20.36 \pm 4.12$ | $20.93 \pm 3.95$ | $11.84 \pm 3.93$ |
| | BWD | $11.27 \pm 2.80$ | $10.91 \pm 2.81$ | $8.97 \pm 2.76$ | $3.33 \pm 2.17$ |



**Table 6. Same as Table 4 but for the FDAs models represented by aggregates of Gaussian random spheres (FDA_GS) or aggregates of droxtals (FDA_DX).**

| Difference in $g$ (%) | $t = 0.0$ | $t = 0.1$ | $t = 0.2$ | $t = 0.3$ |
|---|---|---|---|---|
| FDA_GS | $4.87 \pm 0.47$ | $4.85 \pm 0.47$ | $4.91 \pm 0.47$ | $5.09 \pm 0.47$ |
| FDA_DX | $2.74 \pm 0.61$ | $2.75 \pm 0.61$ | $2.70 \pm 0.62$ | $2.31 \pm 0.62$ |

### 4.3 Habit mixture model

Different morphological features of FDAs were reported by Stith et al. (2014), with the FDAs comprising of mixtures of quasi-spherical FDs and faceted ice crystals (see Fig. 14 in Stith et al. (2014)). The faceted ice crystals within these aggregates often manifested as hexagonal plates or column crystals. However, the CPI instrument used for capturing images of ice crystals is limited in its ability to fully describe the three-dimensional structures of ice crystals. When imaged by the CPI, ice crystals are presented at specific angles, suggesting that what were identified as hexagonal plates in FDAs may actually be hexagonal

facets of droxtals. This interpretation is supported by the experimental findings of Yamazaki and Gonda (1984) and Pedernera and Ávila (2018), which showed that FDs can evolve into hexagonal column-like structures over time.

In previous research, habit mixture models were used to interpret the single-scattering properties of FDAs observed in DCCs during CIRCLE−2. However, the computations of these properties were made indirectly through the weighted habit mixture model (Baran et al., 2012). Baran et al. (2012) introduced four different shape models: Chebyshev particles, spheroids (e.g.,

prolate and oblate), ice spheres, and highly randomized ten-element hexagonal column aggregates. In the paper, weighting applied to each $P_{11}$ of these models and the sum of weighting equals to one. A combination of two different types of Chebyshev particles and a highly irregular ten-element column aggregate with $t = 0.8$, called model 4 in the paper, provided the best fit to the in situ measurements obtained during CIRCLE−2 (see Fig. 8 of Baran et al., 2012).

In this study, the theoretical method from the previous work is advanced by implementing a direct computation approach.

This approach used habit mixture models, constructed by rearranging components of Gaussian random sphere aggregates and droxtal aggregates. These aggregates models were found to minimize the RMSE in comparison with in-situ measurements, as detailed in Sect. 4.2. The single-scattering properties of both the habit mixture models developed in this study and the weighted habit mixture models from previous work were computed under conditions identical to those used for the FDs and FDAs models.

Figure 15 shows the shapes of three different types of habit mixture models (right most column) developed in this study and the corresponding $P_{11}$ and $g$ of them. The first type of models, henceforth referred to as habit mixture type 1 (HM_T1, see top right panel in Fig. 15), consists of aggregates of Gaussian random spheres and droxtals, each demonstrating the minimum RMSE as discussed in Sections 4.1 and 4.2. The second type, called HM_T2 (see middle right panel in Fig. 15), serves as an alternative version of HM_T1, replacing its components with ones exhibiting maximum sphericity, retaining the same number

of components. The third type of habit mixture (HM_T3) as shown in the bottom right panel shares the same shape of Gaussian random spheres as HM_T2, but its droxtal shapes are identical to those in HM_T1. In addition, to investigate the effect of component shape, one Gaussian random sphere was replaced with a droxtal (see bottom right panel in Fig. 15). The 15 different models for each type of habit mixture, or 45 habit mixture models, were generated by varying their 3D morphologies (i.e., *AI*) as discussed in Sect. 2.3. For HM_T1 (HM_T2; HM_T3), the *AI* values range from a minimum of 0.4808 (0.5537; 0.5160) to

a maximum of 0.9502 (0.9729; 0.9787), with a mean value of 0.7421 (0.7654; 0.7544).

A comparison of the single-scattering properties of the newly developed habit mixture models with the observational data from CIRCLE−2 (i.e., filled red circles in the panels; Gayet et al., 2012) and with model 4 of Baran et al. (2012) (i.e., blue dashed line in the panels) is shown in Fig. 15. The first row of Fig. 15 shows the $P_{11}$ of HM_T1 compared against the in situ



measurements. Each column, from left to right, corresponds to $t = 0.0, 0.1, 0.2,$ and 0.3. For the HM_T1, on average, the energy scattered in the forward, lateral, and backward scattering regions for HM_T1 differs by 5.95%, 37.47%, and 9.07% from those of the in situ measurements, respectively. The RMSE was minimized to a value of 0.0713 with $t = 0.3$. The lateral scattering region exhibits slightly improved agreement with the in situ measurements in comparison to the FDAs models which are composed of Gaussian random spheres. However, HM_T1 did not perform as well as those FDA models consisting of homogeneous droxtal components. The HM_T1 models with $t = 0.3$ exhibit high accuracy in simulating the observed $g$, showing an average value of 0.78.

For HM_T2, the comparison between the computed $P_{11}$ and the in situ measured $P_{11}$ is shown in the second row of Fig. 15. The HM_T2 has its minimum RMSE value of 0.1293 with $t = 0.3$. In the forward, lateral, and backward scattering regions, the average differences between HM_T2 and the in situ measurements were 6.53%, 38.84%, and 4.51%, respectively. Compared to HM_T1, HM_T2 more accurately predicts the peaks in the backward scattering region (138° ~ 141°). However, a higher average difference in the lateral scattering region from in situ measurements than those computed for HM_T1 was present. This difference found in the lateral scattering region is attributed to the Gaussian random spheres which are close to spherical shape. The average computed $g$ for HM_T2 was found to be 0.82, which falls within the upper range of the uncertainty of the PN instrument. This calculated $g$ was higher than that of HM_T1, attributed to greater difference in the forward scattering region of HM_T2. However, in the backward scattering region, specifically between 120° and 150°, HM_T2 demonstrated reasonable agreement.

Each panel in the third row of Fig. 15 shows the computed $P_{11}$ of HM_T3. The average differences in the forward, lateral, and backward scattering regions compared to in situ measurements were 1.03%, 2.69%, and 4.33%, respectively. Notably, RMSE for these models minimized to a value of 0.0427 with $t = 0.3$, indicating significant improvement in simulating the in situ measured $P_{11}$, particularly in the lateral scattering regions compared to other habit mixture models. For example, HM_T3 with $t = 0.3$ shows 0.87%, 0.89%, and 5.37% difference in the forward, lateral, and backward scattering regions, respectively. Moreover, it captures the peak in the back-scattering direction more accurately than other habit mixture models. The average $g$ calculated for HM_T3 was 0.81, falling within the uncertainty range estimated by the PN. This value is slightly lower than that of HM_T2 but higher than HM_T1.

For the model 4 developed by Baran et al. (2012) denoted as AB12 in Fig. 15, the average differences of the $P_{11}$ with the in situ measurements were 2.08%, 9.45%, and 12.09% in the forward, lateral, and backward scattering regions respectively. In contrast, HM_T3 with $t = 0.3$ better fits the in situ measurements, showing differences of 0.87%, 0.88% and 5.37% in the forward, lateral, and backward scattering angles, respectively. Consequently, the RMSE for AB12, calculated using Eq. 10, was 0.0663, which is higher than that of HM_T3 with $t = 0.3$ (i.e., 0.0427). Although the average $g$ (0.80 ± 0.01) of HM_T3 was slightly higher than that of the in situ measurements, it still remains within the range observed by the PN (0.78 ± 0.04).



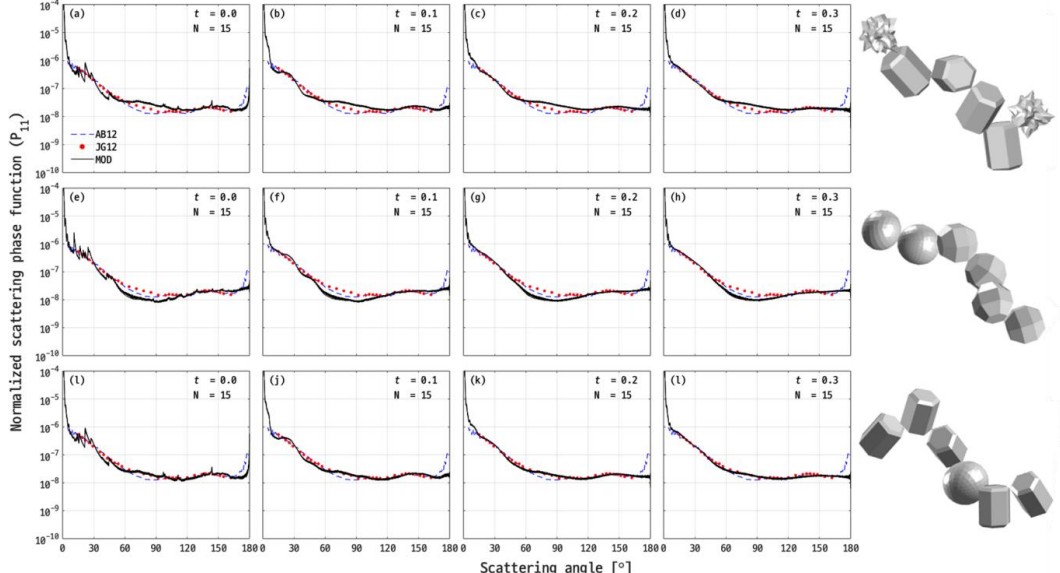

**Figure 15. A comparison of $P_{11}$ between habit mixture models (black solid lines), the weighted habit mixture model developed by Baran et al. (2012) (blue dashed line), and in situ measurements (filled red circles). The $t$ values are displayed in the upper-right corner of each panel. Notations, AB12, JG12, and MOD represent the $P_{11}$ of best-fit weighted habit mixture model from Baran et al. (2012), in situ measured $P_{11}$ from CIRCLE−2 (Gayet et al., 2012), and the calculated $P_{11}$ of habit mixture models developed in this study, respectively. The example images of habit mixture models corresponding to the displayed $P_{11}$ in each row of the figure are shown on the right-most side.**

## 5 Summary and Conclusions

In this study, idealized models representing the shapes of FDs and FDAs observed in deep convective clouds were developed based on in situ aircraft measurements and laboratory experiments, and the corresponding single-scattering properties (i.e., scattering-phase function $P_{11}$ and the asymmetry parameter $g$) were calculated. These computed values of $P_{11}$ and $g$ were subsequently compared with those obtained during the CIRCLE−2 field campaign. Gaussian random spheres and droxtals were proposed as possible candidates for representing the forms of observed FDs and FDAs. A total of 120 individual models of Gaussian random spheres and 129 models of droxtals were generated by varying their shapes. Additionally, by attaching the individual models in both a homogeneous or heterogeneous manner and considering 3D morphologies with varying *AI*, three different types and a total of 315 models of FDAs were created as follows: (1) aggregates of Gaussian random spheres; (2) aggregates of droxtals; and (3) mixtures of Gaussian random spheres and droxtals, which are referred to as habit mixtures. The habit mixture models were further categorized into three different types based on variations in shape and component fractions.

The $P_{11}$ and $g$ of the newly developed models were calculated using a parallelized version of geometric optics method at λ = 0.80 μm for the comparison with the PN measurements from CIRCLE−2. The distortion parameter (*t*), an indirect method to represent the distorted faces, surface roughness, or inclusions of natural ice crystals, was also considered in the calculations. The computed $P_{11}$ was divided into three different scattering regions: forward scattering (0° ~ 60°), lateral scattering (60° ~ 120°), and backward scattering (120° ~ 180°), with differences between the calculations and PN measurements analyzed across these regions. The variation in the $g$ from the calculations and RMSE compared to the PN measurements were also evaluated. The most important findings of this research are summarized below:



1.  Overall, discrepancies over the scattering angles and the RMSE values were minimized when $t = 0.3$ was applied to the shape models developed in this study.

2.  For $P_{11}$ of a single droxtal with $t = 0.3$, differences with the in situ measurement were 2.49%, 3.48%, and 10.64% in the forward, lateral, and backward scattering regions, respectively. The RMSE was minimized to a value of 0.0806. The $g$ value was 0.81 which is close to the upper range of uncertainty observed by the PN ($0.78 \pm 0.04$).

3.  The Gaussian random sphere with $t = 0.3$ was less effective than the droxtal in simulating the in situ measured $P_{11}$, with differences of 6.89%, 39.32%, and 20.16% in the forward, lateral, and backward scattering regions, respectively. These differences, especially in the lateral scattering direction, were one of the characteristics for spherical shape models that contributed to a higher RMSE of 0.1243. Nonetheless, the calculated $g$ of 0.76 fell within the measurement range of PN.

4.  For individual models, despite the calculated $g$ for both droxtals and Gaussian random spheres falling within the measurement range of the PN, the droxtal exhibits better agreement with the in situ measured $P_{11}$ than the Gaussian random sphere. This finding suggests that faceted ice crystals (e.g., on the droxtal) may be responsible for scattered energy into the lateral scattering region of the observations from CIRCLE−2. However, it may be particularly true for the specific cases from CIRCLE−2. Therefore, it is necessary to carefully examine whether faceted ice crystals are a common occurrence in the upper parts of anvil clouds associated with continental DCCs.

5.  For the FDAs with homogeneous components (i.e., either aggregates of Gaussian random spheres or aggregates of droxtals), the aggregates of droxtals with $t = 0.3$ show the minimum RMSE of 0.0549, showing differences from in situ measurements of 1.19%, 12.43%, and 1.41% in the forward, lateral, and backward scattering regions, respectively. The computed $g$ value was found to be 0.79.

6.  In contrast, aggregates of Gaussian random spheres with $t = 0.3$ show larger differences of 10.44%, 67.60%, and 0.27% in the forward, lateral, and backward scattering regions, respectively, leading to a higher RMSE value of 0.1338. The calculated $g$ was 0.75.

7.  The third type habit mixture (i.e., HM_T3), which is a combination of five droxtals and one Gaussian random sphere, matched the PN measurements most closely. It effectively minimized the differences from in situ measurements to 0.87%, 0.89%, and 5.37% in the forward, lateral, and backward scattering directions, respectively. The RMSE for HM_T3 was reduced to a value of 0.0427, and the corresponding $g$ was 0.80.

8.  In the comparison, the difference and RMSE between the weighted habit mixture model developed by Baran et al. (2012) and in situ measurements had larger differences of 2.08%, 9.45%, and 12.09% in the forward, lateral, and backward scattering region, respectively. This resulted in an increased RMSE of 0.0663. The corresponding $g$ was 0.79.

9.  The assumption that FDAs consist of homogeneous components was found to be inadequate for interpreting the in situ measured single-scattering properties. Instead, constructing FDAs from a heterogeneous mixture of quasi-spherical (i.e., Gaussian random spheres) and faceted (i.e., droxtal) components showed the best match for the single-scattering properties of FDAs measured in situ.

The findings of this study have significant implications for improving the accuracy of simulations regarding the radiative impacts of deep convective clouds and associated anvils on the Earth's climate system. A comprehensive understanding of the single-scattering properties of the constituent particles of deep convective clouds is essential to effectively interpret and represent the role of these clouds in large-scale climate models. To this end, in-situ measurements of single-scattering properties obtained during the field campaign were analyzed using shape models that represent the observed habits, along with methods for calculating these properties. In this research, interpreting in situ measured single-scattering properties using models of aggregates with heterogeneous components has proven to be more accurate than aggregate models with homogeneous components or individual shape models. This result agrees with the interpretations with weighted habit mixture models proposed by Baran et al. (2012). Nonetheless, it is important to note that these conclusions may be particularly relevant



to the specific cases analyzed in this study, highlighting the need to investigate whether FDAs with heterogeneous components are a common feature of deep convective clouds (Stith et al., 2014). Furthermore, the in situ measurements used to assess the developed models did not sufficiently resolve the backward scattering region, which are sensitive to variations in the

orientation, shape, and heterogeneities (e.g., distorted surfaces, surface roughness, or inclusions) of ice crystals. In this respect, it should be emphasized that the measurements of the intensities of scattered light across the full range of scattering angles, coupled with images of ice crystals are required. The use of an advanced cloud probe, such as a particle habit imaging and polar scattering probe (Abdelmonem et al., 2016; Schnaiter et al., 2018; Waitz et al., 2021), capable of capturing the detailed 3D morphologies of FDs or FDAs, is essential to further this understanding. Additionally, cloud chamber experiments, where

the atmospheric conditions of deep convective clouds can be accurately replicated and studied repeatedly, could offer a viable solution for obtaining more accurate single-scattering measurements or exploring the mechanisms of producing FDAs which is not covered in this study.

**Code Availability**

Code is available from the corresponding author on reasonable request.

**Data Availability**

All raw data can be provided by the corresponding authors upon reasonable request.

**Author Contributions**

JK, JU, and GMM conceived the study. JK and SP ran the model simulations. AJB analyzed and provided CIRCLE−2 PN
measurements. JK, SP, JU, JWC, KL, SSL, CHJ, KSL performed the result comparison. JK prepared the manuscript with contributions from all co-authors. All authors were involved in the scientific interpretation and discussion.

**Competing Interests**

One author is a member of the editorial board of Atmospheric Chemistry and Physics

**Acknowledgements**

This work was supported by 2023 BK21 FOUR Graduate School Innovation Support funded by Pusan National University (PNU-Fellowship program). This work was also supported by the National Research Foundation of Korea (NRF) grant funded by the Korea government (MSIT) (NRF-2020R1A2C1013278 and NRF-2023R1A2C1002367), by Basic Science Research Program through the NRF funded by the Ministry of Education (NRF-2020R1A6A1A03044834), and by the United States National Science Foundation Award 1842094. This work was funded by the Korea Meteorological Administration Research

and Development Program "Research on Weather Modification and Cloud Physics" under Grant (KMA2018-00224). We thank A. Macke for the original ray-tracing code and K. Muinonen and T. Nousiainen for the Gaussian random sphere SIRIS code and Jeffrey Stith for useful suggestions regarding this work while one of us (GM) was on sabbatical at NSF NCAR. This study was conducted employing the system of Korea Meteorological Institute, communally used by weather and climate industry.




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
