# Peer review of "On the calculation of single-scattering properties of frozen droplets and frozen droplet aggregates observed in deep convective clouds"

_EGUsphere, 2024_

## Author Comment (AC1)

**RC1**

This study presents model-data comparisons on the scattering phase function of frozen-droplet and frozen-droplet aggregates in deep convective clouds. The data are obtained from the CIRCLE−2 field campaign. Gaussian random spheres and droxtals were proposed as possible candidates for representing the forms of observed frozen droplets and frozen droplet aggregates. The authors generate a total of 120 individual models of Gaussian random spheres and 129 models of droxtals and by attaching the individual models in both a homogeneous and heterogeneous manner, a total of 315 models of three different types of droplet aggregates model were generated. This study has some interesting findings, such as the differences between faceted particle and quasi-spherical particles on the phase function comparison. Given that the study provides quite many comparisons on different types of models, this study might have some reference value on this topic.

We sincerely thank the reviewer for the careful reading of the manuscript and the many helpful and constructive suggestions that have improved the quality of the manuscript substantially.

In response to the reviewers' comments, we have revised the manuscript, with the major modifications outlined as follows. First, we extended the calculations by increasing the distortion parameter ($t$) to a maximum of 0.95 in increments of 0.05, resulting in changes to Sections 4.1, 4.2, and 4.3. The analysis of these extended results has been incorporated into the revised manuscript. Additionally, as the reviewers pointed out, to avoid redundancy with Section 4.1, we have removed Sections 4.2.1 and 4.2.2, which compared results for homogeneous component aggregates represented by Gaussian random spheres or droxtals, from the main text and included them in the Supplement (S1 and S2). Another significant modification is the introduction of an additional criterion: the number of angles ($\omega$) falling within the ±20% uncertainty range of PN, which was added to assess the accuracy of our theoretical calculations against observational data. Further discussion on $\omega$ has been integrated into the revised manuscript. Lastly, we modified the method used to construct habit mixture models, as explained in Section 4.3, and the corresponding comparison results are now thoroughly discussed in that section. Although significant revisions have been made in this study, the core results presented in the original submission remain unchanged, demonstrating the robustness of our findings.

**Specific comments:**

1. What are relative measurement uncertainties in each region of scattering angle?

The following paragraph about the operation and measurement uncertainty of PN is added in Section 4.

"*The PN, as detailed by Gayet et al. (1997), is an airborne instrument designed to measure the angular scattering pattern, or scattering phase function, of an ensemble of cloud particles ranging from a few micrometers to about 1 mm in diameter. Operating at a wavelength of 0.8 µm, the PN captures scattering angles between ±15° and ±162° with a resolution of 3.5°, typically providing data at 32 distinct angles from among 56 photodiodes (Jourdan et al., 2010). Measurements at near-forward and backward angles (θ < 15° and θ > 162°) are less reliable due to diffraction effects caused by the edges of holes drilled in the paraboloidal mirror (Gayet et al., 1997). To ensure continuous sampling, the PN integrates the signals from each photodiode over periods selectable by the operator, commonly around 100 ms. The average measurement errors for the angular scattering coefficients range from 3% to 5% for angles between 15° and 162°, with a maximum error reaching 20% at the outermost angles (Shcherbakov et al., 2006). The instrument's ability to directly measure the scattering phase function allows for differentiation of particle types and calculation of essential optical parameters, such as the extinction coefficient and g. Gayet et al. (2002) reported an uncertainty of 25% for the PN-derived extinction coefficient, while the estimated absolute error for the g ranges from ±0.04 to ±0.05, depending on the prevalence of large ice crystals within the cloud (Jourdan et al., 2010).*".

2. Could you provide some in-situ images for the support of these idealized shape models?

Example images of frozen droplets and their chain-shaped aggregates captured by the Cloud Particle Imager (CPI) during field campaigns are shown below. As stated in the manuscript, although a high-resolution CPI (i.e., 2.3 µm) was used to image the frozen droplets, its resolution was not sufficiently high to fully resolve their three-dimensional morphological features.

Example CPI images of frozen droplets and their chain-shaped aggregates, sampled during the DC3 field campaign (Um et al., 2018), are shown below.

[Figure]

**Figure 1.** (a) Example CPI images of ice crystals observed at $T = -58.16\,°C$ (altitude of 12.11 km) between 22:12:13 and 22:12:19 UTC, and (b) example CPI images of ice crystals observed at $T = -57.72\,°C$ (altitude of 12.03 km) between 22:21:02 and 22:22:14 UTC. The 200 µm scale bar is embedded in each figure.

Example of CPI images from the CIRCLE-2 field campaign (Gayet et al., 2012) are also shown below.

[Figure]

**Fig. 10.** Typical examples of chain-like aggregates ice crystals from 2 up to 15 individual frozen droplets.

3. The model only considers surface scattering effects and aggregate configurations, what about internal scattering?

We did not consider the effect of any internal inclusions in this study. Multiple previous studies (Macke et al. 1996; Labonnote et al. 2001; Xie et al. 2009; Bi and Yang 2014; Panetta et al. 2016; Smith et al. 2016) have shown similar effects, i.e., featureless $P_{11}$ and lower $g$, as the surface effects.

Bi, L. and P. Yang: Accurate simulation of the optical properties of atmospheric ice crystals with the invariant imbedding T–matrix method, J. Quant. Spectrosc. Radiat. Transf., 138, 17–35, 2014.

Macke, A., Mishchenko, M. I., and Cairns, B.: The influence of inclusions on light scattering by large ice particles, J. Geophys. Res., 101, 23311-23316, 1996.

Labonnote, L. C., Brogniez, G., Buriez, J. C., Doutriaux–Boucher, M., Gayet, J.-F., and Macke, A.: Polarized light scattering by inhomogeneous hexagonal mono crystals: Validation with ADEOS–POLDER measurements, J. Geophys. Res., 106, 12139–12155, 2001.

Panetta, R. L., J.-N. Zhang, L. Bi, P. Yang, and G. Tang: Light scattering by hexagonal ice crystals with distributed inclusions. J. Quant. Spect. Radiative Transf., 178, 336-349, 2016.

Smith, H. R., A. J. Baran, E. Hesse, P. G. Hill, P. J. Connolly, and A. Webb: Using laboratory and field measurements to constrain a single habit shortwave optical parameterization for cirrus, Atmos. Res., 180, 226-240, 2016.

Xie, Y., Yang, P., Kattawar, G. W., Minnis, P., and Hu, Y. X.: Effect of the inhomogeneity of ice crystals on retrieving ice cloud optical thickness and effective particle size, J. Geophys. Res., 114, D11203, doi:10.1029/2008JD011216, 2009.

4. Why is the distortion parameter only set up to 0.3? Since high distortion provides the best fit, presumably higher distortion could give even better fit. It is suggested to increase the distortion up to at least 0.6 and perform the comparison again.

Following the reviewers' comment, we have extended the calculations with the distortion parameter increased up to 0.95 and the analysis of the results has been included in the revised manuscript. The following paragraph is added in Section 5.

"*The application of the distortion parameter (t) in the calculations of single-scattering properties facilitated the production of smoother $P_{11}$, reduces g, and increases ω, thereby enhancing the agreement with the PN measurements to certain extent (e.g., up to t = ~0.3). However, beyond this threshold, larger t values diminished the agreement.*"

5. It is suggested to have more discussion about the differences among the three different regions, that is, forward, lateral, backward scattering. How these angular regions related to the particle shapes.

Following the reviewer's comment, additional discussions have been incorporated throughout the manuscript.

6. It would be better reducing the number of significant findings to highlight the most important ones in the conclusion section.

Following the reviewers' comment, the number of significant findings stated in Section 5 Summary and Conclusions have been reduced without losing the importance of the findings.

7. It may be more useful to not just state the relative differences, but also state the signs of relative differences in three different scattering regions.

Following the reviewer's comment, the signs of the relative differences have been added throughout the manuscript.

---

## Author Comment (AC2)

**RC3**

This study aims to find optical models for small ice crystals identified as frozen drops or frozen drops aggregates. Several optical models are compared to data obtained during the CIRCLE-2 campaign. The manuscript is well written and the topic is of interest for ACP. However, I have several questions that should be addressed before I would recommend publication.

We sincerely thank the reviewer for the careful reading of the manuscript and the many helpful and constructive suggestions that have improved the quality of the manuscript substantially.

In response to the reviewers' comments, we have revised the manuscript, with the major modifications outlined as follows. First, we extended the calculations by increasing the distortion parameter ($t$) to a maximum of 0.95 in increments of 0.05, resulting in changes to Sections 4.1, 4.2, and 4.3. The analysis of these extended results has been incorporated into the revised manuscript. Additionally, as the reviewers pointed out, to avoid redundancy with Section 4.1, we have removed Sections 4.2.1 and 4.2.2, which compared results for homogeneous component aggregates represented by Gaussian random spheres or droxtals, from the main text and included them in the Supplement (S1 and S2). Another significant modification is the introduction of an additional criterion: the number of angles ($\omega$) falling within the ±20% uncertainty range of PN, which was added to assess the accuracy of our theoretical calculations against observational data. Further discussion on $\omega$ has been integrated into the revised manuscript. Lastly, we modified the method used to construct habit mixture models, as explained in Section 4.3, and the corresponding comparison results are now thoroughly discussed in that section. Although significant revisions have been made in this study, the core results presented in the original submission remain unchanged, demonstrating the robustness of our findings.

- The observations are an important part of this paper but they are hardly described at all. More details need to added. Specific questions are:
    1. What does the PN measure and how?
    2. Are these single crystal or bulk observations?
    3. What selection of data is made? Are there only FD or FDA's in this sample?
    4. What do the grey ranges in figure 5 and others mean? Are the spikes seen in these ranges real features or more like noise?
    5. How can we interpret the mean of the observations that is used as a target?

To better interpret PN measurements and remove ambiguity, we depict only the PN-measured average $P_{11}$ (i.e., black filled circles) of frozen droplet aggregates, as presented in Gayet et al.

(2012) and Baran et al. (2012), in the new figures of this manuscript. The gray-shaded areas shown in the original figures, representing the full range of $P_{11}$ measurements obtained during the entire CIRCLE-2 campaign, have been removed in the new figures. Additional detailed information about PN measurements has been added.

At the beginning of Section 4, the following sentences have been included:

*"The PN, as detailed by Gayet et al. (1997), is an airborne instrument designed to measure the angular scattering pattern, or scattering phase function, of an ensemble of cloud particles ranging from a few micrometers to about 1 mm in diameter. Operating at a wavelength of 0.8 μm, the PN captures scattering angles between ±15° and ±162° with a resolution of 3.5°, typically providing data at 32 distinct angles from among 56 photodiodes (Jourdan et al., 2010). Measurements at near-forward and backward angles (θ < 15° and θ > 162°) are less reliable due to diffraction effects caused by the edges of holes drilled in the paraboloidal mirror (Gayet et al., 1997). To ensure continuous sampling, the PN integrates the signals from each photodiode over periods selectable by the operator, commonly around 100 ms. The average measurement errors for the angular scattering coefficients range from 3% to 5% for angles between 15° and 162°, with a maximum error reaching 20% at the outermost angles (Shcherbakov et al., 2006). The instrument's ability to directly measure the scattering phase function allows for differentiation of particle types and calculation of essential optical parameters, such as the extinction coefficient and g. Gayet et al. (2002) reported an uncertainty of 25% for the PN-derived extinction coefficient, while the estimated absolute error for the g ranges from ±0.04 to ±0.05, depending on the prevalence of large ice crystals within the cloud (Jourdan et al., 2010)."*.

At the beginning of Section 4.1.1, the following sentence has been revised:

*"Figure 5a illustrates the comparison between the $P_{11}$ of 120 single FDs models, represented by Gaussian random spheres, and the PN measured average $P_{11}$ (Baran et al., 2012; Gayet et al., 2012) obtained in the developed overshooting convective cell at 11,080 m altitude (T=−58 °C) at 13:08:15–13:08:40 UTC on 26 May, 2007 during CIRCLE−2 (i.e., black filled circles)."*.

Measurements of FD and FDAs were confirmed with CPI images during the CIRCLE-2 campaign. Gayet et al. (2012) stated, *"A visual classification roughly gives a proportion of 70 % of typical chains of ice crystals and ice particles exhibiting a faceted shape have been rarely observed."*. Because of this reason, we emphasize the need for coupled measurements of light angular intensity and image of the same particle in Section 5 – *"In this respect, it should be emphasized that the measurements of the intensities of scattered light across the full range of*

*scattering angles, coupled with images of ice crystals are required. The use of an advanced cloud probe, such as a particle habit imaging and polar scattering probe (Abdelmonem et al., 2016; Schnaiter et al., 2018; Waitz et al., 2021), capable of capturing the detailed 3D morphologies of FDs or FDAs, is essential to further this understanding.".*

- Related to the last question about the data: Why is the mean of the observations an appropriate target of an optical model? Should the variation not be represented by a set of models? On line 598 it is stated that "the assumption that FDAs consist of homogeneous components was found to be inadequate for interpreting the in situ measured single-scattering properties." What is the criterion for calling these other models inadequate? They are close to the mean of the observations and within the grey range. So how well should any model fit the mean of the observations to be deemed an adequate model?

We fully agree with the reviewer's concerns regarding the natural variabilities of the optical properties of ice crystals. It is indeed challenging to mimic every detailed optical feature of natural ice crystals in ice clouds using idealized optical models. Many factors, such as size, shape, surface roughness, inclusions, and orientation, contribute to this variability. Additionally, there are observational uncertainties when measuring the optical properties of ice crystals.

The PN instrument, like other instruments, has inherent uncertainties, and the resulting measurements reflect these uncertainties. Despite these uncertainties, it is necessary to evaluate the accuracy of our theoretical calculations against observations. To do this, we need a specific criterion. In this study, we selected the root mean square error (RMSE) between the theoretical calculations and the PN-measured average $P_{11}$ and newly introduced parameter, the number of angles ($\omega$) falling within the ±20% uncertainty range of PN, as our criteria.

- The effect of distortion on the models is investigated. The best match is found for the highest distortion applied. Therefore, I suggest to also apply higher distortion values to show that the optimum is indeed at 0.3, or whether it is at a higher value. Also please indicate for which specific case of Gaussian random sphere and which specific droxtal type the distortion is applied. I also suggest adding an extra panel just as in Fig 6, 8, 11 and 14 showing the change in g as a function of t.

In response to the reviewer's comment, we have extended the calculations by increasing the distortion parameter $t$ to a maximum of 0.95, and the analysis of these results has been included in the revised manuscript. We have also made additional revisions to specify the cases for the Gaussian random sphere and droxtal types where the distortion is applied. Furthermore, we have added an extra panel, as suggested, showing the change in $g$ as a function of $t$ in the revised manuscript.

- On line 603 you state that "the findings of this study have significant implications for improving the accuracy of simulations regarding the radiative impacts of deep convective clouds and associated anvils on the Earth's climate system". You did not show this. What is the basis for this statement? Often optical models with smooth phase functions and asymmetry parameters close to those you are finding are used in such calculations, so I would not expect a large impact. Please discuss a firm basis for this statement or remove or weaken this statement, also in the abstract.

The sentence on line 603, "*The findings of this study have significant implications for improving the accuracy of simulations regarding the radiative impacts of deep convective clouds and associated anvils on the Earth's climate system.*", has been replaced with, "*The findings of this study suggest potential implications for improving the accuracy of simulations regarding the radiative impacts of deep convective clouds and associated anvils on the Earth's climate system.*".

Additionally, the sentence in the abstract, "*The result of this study carries important implications for enhancing the calculation of single-scattering properties of DCCs.*", has been replaced with, "*The results of this study suggest potential implications for enhancing the calculation of single-scattering properties of ice crystals in DCCs.*".

- In the conclusions, the findings numbered 2, 3, 5, 6, 7, 8 are too detailed in my opinion and I suggest removing them.

Following the reviewers' comment, the number of significant findings stated in Section 5 Summary and Conclusions have been reduced without losing the importance of the findings.

---

## Author Comment (AC3)

**RC2**

**General comments:**

The authors provide a conclusive and comprehensible assessment of the scattering behavior of small quasi-spherical ice particles in the cirrus anvil region of high convective clouds. They vary the shapes of individual particles and their aggregates over a large range, calculate their scattering properties on the basis of geometrical optics and find an optimal constellation by minimizing the difference between modelled and observed scattering behaviour. Interestingly, the authors find very good agreement with previous work (Baran et al. 2012), which argues for a universal scattering behavior of this particle type. The introduction to the topic and the description of the methodology are very good. I expressly endorse the publication. However, large parts are rather a kind of painstaking work due to the very repetitive procedure for the different crystal types and their aggregates. I suggest shortening it substantially and not going through every variation including illustrations. Perhaps one could only occasionally refer to the results.

We sincerely thank the reviewer for the careful reading of the manuscript and the many helpful and constructive suggestions that have improved the quality of the manuscript substantially.

In response to the reviewers' comments, we have revised the manuscript, with the major modifications outlined as follows. First, we extended the calculations by increasing the distortion parameter ($t$) to a maximum of 0.95 in increments of 0.05, resulting in changes to Sections 4.1, 4.2, and 4.3. The analysis of these extended results has been incorporated into the revised manuscript. Additionally, as the reviewers pointed out, to avoid redundancy with Section 4.1, we have removed Sections 4.2.1 and 4.2.2, which compared results for homogeneous component aggregates represented by Gaussian random spheres or droxtals, from the main text and included them in the Supplement (S1 and S2). Another significant modification is the introduction of an additional criterion: the number of angles ($\omega$) falling within the ±20% uncertainty range of PN, which was added to assess the accuracy of our theoretical calculations against observational data. Further discussion on $\omega$ has been integrated into the revised manuscript. Lastly, we modified the method used to construct habit mixture models, as explained in Section 4.3, and the corresponding comparison results are now thoroughly discussed in that section. Although significant revisions have been made in this study, the core results presented in the original submission remain unchanged, demonstrating the robustness of our findings.

The use of aggregates consisting of identical particles could be omitted, as the scattering properties of both do not differ that much, as the authors themselves show, and which is also known from the literature.

Following the reviewer's comment, Sections 4.2.1 and 4.2.2 have been omitted from the revised manuscript, but we have included them along with the corresponding figures and tables in the Supplement (S1 and S2).

I would be very interested to know whether the results at the end of the mixed particle aggregates could be obtained by a simple averaging of random Gaussian spheres and doxtrals, i.e. by something like

pf_best(c, t1, t2) = c*pf_rgs(t1) + (1-c)pf_dox(t2)

with a fraction c and two optimal distortions t1 and t2 for the Gaussian random spheres and the doxtrals, respectively. This could also be shown in Fig. 15.

In fact, you could shorten section 4.1 and 4.2 substantially and focus more on 4.3.

In response to the reviewer's comment, we conducted additional simulations using weighted habit mixture models. This indirectly generated habit mixture model was less effective in simulated the in situ measured single-scattering properties compare to the models that directly aggregate components (i.e., direct method). The results of these simulations have been included in the Supplement (S3).

Another concern of mine is that the PN measurements do not cover the full range of forward scattering. And because of the very strong forward scattering behaviors, the measurements miss let's say 99% of the scattered energy. So, does it make sense to tune to observations that only cover 1% of the scattered energy?

We agree with your concern. This is a known limitation of current observational capabilities. At present, no instrument can measure scattered light intensity across the full range of scattering angles. We have acknowledged this limitation and emphasized the need for the development of new instruments to address this issue at the end of the manuscript.

**Specific comments:**

line 24: "cloud radiative forcing" -> "cloud radiative effect"

Done. The term "cloud radiative forcing" has been changed to "cloud radiative effect".

l 51 - 52: One could argue that the radiative effects of deep clouds are to a certain extent "saturated" due to the asymptotic behavior of the radiative fluxes with increasing optical thickness. Therefore, subtle changes in scattering properties may *not* play an important role for this cloud type.

We agree with the reviewer's comment. However, we have also observed the occurrence of frozen droplets and their aggregates in the extensive outflow regions of anvil clouds. The optical depth of these outflow regions is generally much smaller than that of the convective towers of anvil clouds. Therefore, we believe that the scattering properties of ice crystals, particularly for radiance applications, remain significant.

l 72 - 74: As projected area is radiatively more relevant than number concentration, this means that more than half of the scattering is not by FDs and FDAs! Do you know the shape of those particles?

Thank you very much for your question. Upon revisiting Um et al. (2018), we identified typographical errors in the study. The fractions of frozen droplets (FDs) and their aggregates (FDAs) were 73.036% and 20.850% by number, respectively, while they were 40.014% and 46.308% by projected area, respectively. Therefore, lines 72–74 should be corrected to: "*It was revealed that FDs and FDAs were the predominant habits, comprising 93.9% (by number) and 86.3% (by projected area) of the observed particles, respectively.*" This correction has been made in the revised manuscript. Other observed habits included plates, columns, and unclassifiable crystals, as shown in Table 1 of Um et al. (2018), which is attached below.

**Table 1.** Segregated time periods of the 6 June flight and contributions (%) of crystal habit to the total number (total projected area) of ice crystals for the given time period. The average and standard deviation of temperature ($T$), altitude, and maximum dimension ($D_{max}$) of ice crystals determined from CPI images are also listed for the given time period.

| Period | Time (UTC) | $D_{max}$ (µm), $T$ (°C), altitude (km) | Single frozen droplet | Frozen droplet aggregates (FDAs) | Plate (PLT) | Column (COL) | Unclassified (UC) |
|---|---|---|---|---|---|---|---|
| All | 22:11:00–22:28:00 | $80.7 \pm 45.4$, $-58.1 \pm 1.4$, $12.121 \pm 0.138$ | 73.036 (40.014), $34.4 \pm 6.8$ | 20.850 (46.308), $80.7 \pm 45.4$ | 0.013 (0.059), $98.1 \pm 30.9$ | 0.013 (0.022), $69.6 \pm 12.7$ | 6.073 (13.539), $75.7 \pm 37.2$ |
| 1 | 22:11:00–22:15:00 | $68.4 \pm 37.1$, $-60.0 \pm 1.2$, $12.226 \pm 0.151$ | 84.065 (65.691), $32.5 \pm 5.6$ | 12.050 (27.786), $68.4 \pm 37.1$ | – | – | 3.885 (6.523), $53.1 \pm 23.5$ |
| 2 | 22:19:00–22:23:40 | $72.4 \pm 42.9$, $-57.5 \pm 0.3$, $12.033 \pm 0.003$ | 70.635 (39.354), $34.6 \pm 6.5$ | 24.002 (49.615), $79.4 \pm 42.9$ | – | – | 5.340 (10.922), $73.6 \pm 30.4$ |
| 3 | 22:23:50–22:28:00 | $84.4 \pm 48.8$, $-56.6 \pm 0.5$, $12.032 \pm 0.004$ | 71.236 (35.423), $34.9 \pm 7.4$ | 21.216 (47.467), $84.4 \pm 48.8$ | 0.030 (0.115), $98.1 \pm 30.9$ | 0.030 (0.043), $69.6 \pm 12.7$ | 7.478 (16.922), $81.1 \pm 41.2$ |

l 112 - 114: Sentence duplications from above

The original sentence, "*In this study, shape models representing quasi-spherical FDs and FDAs were developed using Gaussian random spheres and droxtals based on the shapes of these particles observed during field campaigns and laboratory experiments.*", has been replaced with a new sentence, "*In this study, these models were developed using Gaussian random spheres and droxtals, based on the shapes observed during field campaigns and laboratory experiments.*".

l 206: Is this justified by observations? And if so, is the scattering at the aggregate significantly different from that of their individual components?

The sentence, "The shape of all FDs composing a FDA model is identical." has been deleted.

Example images of frozen droplets and their chain-shaped aggregates captured by the Cloud Particle Imager (CPI) during field campaigns are shown below. As stated in the manuscript, although a high-resolution CPI (i.e., 2.3 µm) was used to image the frozen droplets, its resolution was not sufficiently high to fully resolve their three-dimensional morphological features. Thus, it cannot be justified by observations.

Example CPI images of frozen droplets and their chain-shaped aggregates, sampled during the DC field campaign (Um et al., 2018), are shown below.

[Figure]

**Figure 1. (a)** Example CPI images of ice crystals observed at $T = -58.16\,°C$ (altitude of 12.11 km) between 22:12:13 and 22:12:19 UTC, and **(b)** example CPI images of ice crystals observed at $T = -57.72\,°C$ (altitude of 12.03 km) between 22:21:02 and 22:22:14 UTC. The $200\,\mu m$ scale bar is embedded in each figure.

Example of CPI images from the CIRCLE-2 field campaign (Gayet et al., 2012) are also shown below.

[Figure]

**Fig. 10.** Typical examples of chain-like aggregates ice crystals from 2 up to 15 individual frozen droplets.

l 252: are you comparing at discrete angles or for angular intervals?

We are comparing at discrete angles.

l 272: "compared with the PN measurements": The PN show several scattering maxima at about 25, 45 and 55 degrees. Could these be halo features or other indications of hexagonal structures of the ice crystals?

Btw, is the variability of the PN measurements also caused by specific orientations of the particles? Probably not, as they are quasi-spherical. Just curious. And how do you know that the PN does not contain measurements of other particles than FDs and FDAs?

To better interpret PN measurements and remove ambiguity, we depict only the PN-measured average $P_{11}$ (i.e., black filled circles) of frozen droplet aggregates, as presented in Gayet et al. (2012) and Baran et al. (2012), in the new figures of this manuscript. The gray-shaded areas shown in the original figures, representing the full range of $P_{11}$ measurements obtained during the entire CIRCLE-2 campaign, have been removed in the new figures. Additional detailed information about PN measurements has been added.

At the beginning of Section 4, the following sentences have been included:

*"The PN, as detailed by Gayet et al. (1997), is an airborne instrument designed to measure the angular scattering pattern, or scattering phase function, of an ensemble of cloud particles ranging from a few micrometers to about 1 mm in diameter. Operating at a wavelength of 0.8 μm, the PN captures scattering angles between ±15° and ±162° with a resolution of 3.5°, typically providing data at 32 distinct angles from among 56 photodiodes (Jourdan et al., 2010). Measurements at near-forward and backward angles (θ < 15° and θ > 162°) are less reliable due to diffraction effects caused by the edges of holes drilled in the paraboloidal mirror (Gayet et al., 1997). To ensure continuous sampling, the PN integrates the signals from each photodiode over periods selectable by the operator, commonly around 100 ms. The average measurement errors for the angular scattering coefficients range from 3% to 5% for angles between 15° and 162°, with a maximum error reaching 20% at the outermost angles (Shcherbakov et al., 2006). The instrument's ability to directly measure the scattering phase function allows for differentiation of particle types and calculation of essential optical parameters, such as the extinction coefficient and g. Gayet et al. (2002) reported an uncertainty of 25% for the PN-derived extinction coefficient, while the estimated absolute error for the g ranges from ±0.04 to ±0.05, depending on the prevalence of large ice crystals within the cloud (Jourdan et al., 2010)."*.

At the beginning of Section 4.1.1, the following sentence has been revised:

*"Figure 5a illustrates the comparison between the $P_{11}$ of 120 single FDs models, represented by Gaussian random spheres, and the PN measured average $P_{11}$ (Baran et al., 2012; Gayet et al., 2012) obtained in the developed overshooting convective cell at 11,080 m altitude (T=−58 °C) at 13:08:15–13:08:40 UTC on 26 May, 2007 during CIRCLE−2 (i.e., black filled circles)."*.

Since the PN measures the $P_{11}$ of an ensemble of cloud particles, the effect of specific orientation of particles on the measurement would be minimal.

Measurements of FD and FDAs were confirmed with CPI images during the CIRCLE-2 campaign. Gayet et al. (2012) stated, *"A visual classification roughly gives a proportion of 70 %*

*of typical chains of ice crystals and ice particles exhibiting a faceted shape have been rarely observed."*. Because of this reason, we emphasize the need for coupled measurements of light angular intensity and image of the same particle in Section 5 – *"In this respect, it should be emphasized that the measurements of the intensities of scattered light across the full range of scattering angles, coupled with images of ice crystals are required. The use of an advanced cloud probe, such as a particle habit imaging and polar scattering probe (Abdelmonem et al., 2016; Schnaiter et al., 2018; Waitz et al., 2021), capable of capturing the detailed 3D morphologies of FDs or FDAs, is essential to further this understanding."*.

l 285: No, the tilt angle do not mimic surface roughness or inclusions.
The phrase "surface roughness, or inclusions" has been deleted from that sentence.

l 287: azimuth is tilted between 0 and 2pi, zenith between 0 and pi
The sentence has been modified to: *"The zenith and azimuth tilt angles are randomly selected with an equal distribution between 0 and $\theta_t^{max}$ and between 0 and 2π, respectively."*.

l 291 - 292: Fig. 6). "A single Gaussian random sphere with t= 0.3 was the best-fit model...": I have no doubt that this is the case. However, it would be nice to see results for t = 0.4 and 0.5, just to see that the modeled phase function again deviates more from the observations.
Following the reviewers' comment, we have extended the calculations with the distortion parameter increased up to 0.95 and the analysis of the results has been included in the revised manuscript.

l 377: doubling: ". The aggregates of Gaussian random
spheres showed a" -> "showing"
It has been revised to: *"Figure S1 illustrates the $P_{11}$ and g at λ = 0.80 μm for 27 aggregates of Gaussian random spheres (depicted by blue shaded area), which show an average difference of 1.02 ± 0.55, 18.49 ± 5.08%, and 2.86 ± 2.05% from the in-situ measurements in the forward, lateral, and backward scattering regions, respectively."* Also, this sentence has been relocated to S1 in the Supplement.

l 380: "...discrepancies in the lateral scattering angles remain.": Yes, because the scattering of aggregates is close to that of their individual (and identical) components. It is therefore possible that this exercise can be omitted as long as the individual FDs in the aggregate are all identical. See also my general comment above.

Following the reviewer's comment, Sections 4.2.1 and 4.2.2 have been omitted from the revised manuscript, but we have included them along with the corresponding figures and tables in the Supplement.

Figs 10 and 13 could be merged into one figure by using different colors for the different particle shapes.

Figures 10 and 13 have been relocated to the Supplement and are now labeled as Figures S3 and S6, respectively. This change was necessary following the removal of Sections 4.2.1 and 4.2.2 from the revised manuscript. Additionally, we expanded our calculations to include distortion parameters up to 0.95, resulting in a larger number of panels in Figures S3 and S6. Due to this increased complexity, we opted to keep these figures separate rather than merging them into a single figure.

Should a good scattering model not only fit to the mean observations but also to their variability?

Ideally, a good scattering model may fit both the mean observations and their variability. However, this is challenging due to the inherent complexity of ice crystal properties and measurement uncertainties. Our approach, which uses RMSE and $\omega$ as criteria, aims to evaluate accuracy by considering both the mean observations and a reasonable range of variability.

Summary and Conclusions: I suggest to not repeat all the numbers (percentage differences) here but rather to provide a qualitative statement on the (dis)agreements between results from models and observations.

Following the reviewer's comment, a qualitative statement has been added alongside the quantitative statement in Section 5 Summary and Conclusions.

---

## Author Response (AR2)

**Public justification (visible to the public if the article is accepted and published)**: ACP's guidelines for authors request that the article title "highlight the scientific results/findings or implications of the study" whereas less preferred titles highly "only the topic". I wonder if the authors might consider whether a slight revision to the title might be more in line with ACP guidelines, such as revising "On the calculation ..." to "Improved calculation ..."? I defer to the authors' preferences, and am offering this opportunity to optionally revise the title according to ACP guidelines prior to final acceptance or keep it as is. I will then recommendation publication as is.

The title has been revised to "Improved calculation of single-scattering properties of frozen droplets and frozen droplet aggregates observed in deep convective clouds" as recommended.

Thank you to the reviewers and editor.